# FEDERATED LEARNING WITH PARTIAL MODEL PERSONALIZATION

## ABSTRACT

We propose and analyze a general framework of federated learning with partial model personalization. Compared with full model personalization, partial model personalization relies on domain knowledge to select a small portion of the model to personalize, thus imposing a much smaller on-device memory footprint. We propose two federated optimization algorithms for training partially personalized models, where the shared and personal parameters are updated either simultaneously or alternately on each device, but only the shared parameters are communicated and aggregated at the server. We give convergence analyses of both algorithms for minimizing smooth nonconvex functions, providing theoretical support of them for training deep learning models. Our experiments on real-world image and text datasets demonstrate that (a) partial model personalization can obtain most of the benefit of full model personalization with a small fraction of personalized parameters, and, (b) the alternating update algorithm often outperforms the simultaneous update algorithm.

## 1 INTRODUCTION

Federated Learning (McMahan et al., 2017) has emerged as a powerful paradigm for distributed and privacy-preserving machine learning over a large number of edge devices (see Kairouz et al., 2021, and references therein). We consider a typical setting of Federated Learning (FL) with $n$ devices (also called clients), where each device $i$ has a training dataset of $N_i$ samples $z_{i,1}, \cdots, z_{i,N_i}$. Let $w \in \mathbb{R}^d$ represent the parameters of a (supervised) learning model and $f_i(w, z_{i,j})$ be the loss of the model on the training example $z_{i,j}$. Then the loss function associated with device $i$ is $F_i(w) = (1/N_i) \sum_{j=1}^{N_i} f_i(w, z_{i,j})$. A common objective of FL is to find model parameters that minimize the weighted average loss across all devices (without transferring the datasets):

$$\underset{w}{\text{minimize}} \quad \sum_{i=1}^{n} \alpha_i F_i(w), \tag{1}$$

where weights $\alpha_i$ are nonnegative and satisfy $\sum_{i=1}^{n} \alpha_i = 1$. A common practice is to choose the weights as $\alpha_i = N_i/N$ where $N = \sum_{k=1}^{n} N_k$, which corresponds to minimizing the unweighted average loss across all samples from the $n$ devices: $(1/N) \sum_{i=1}^{n} \sum_{j=1}^{N_i} f_i(w, z_{i,j})$.

The main motivation for minimizing the average loss over all devices is to leverage their collective statistical power for better generalization, because the amount of data on each device can be very limited. This is especially important for training modern deep learning models with large number of parameters. However, this argument assumes that the datasets from different devices are sampled from the same, or at least very similar, distributions. Given the diverse characteristics of the users and increasing trend of personalized on-device services, such an i.i.d. assumption may not hold in practice. Thus, the one-model-fits-all formulation in (1) can be less effective and even undesirable.

Several approaches have been proposed for personalized FL, including ones based on multi-task learning (Smith et al., 2017), meta learning (Fallah et al., 2020), and proximal methods (Dinh et al., 2020; Li et al., 2021). A simple formulation that captures their main idea is

$$\underset{w_0, \{w_i\}_{i=1}^{n}}{\text{minimize}} \quad \sum_{i=1}^{n} \alpha_i \Big( F_i(w_i) + \frac{\lambda_i}{2} \|w_i - w_0\|^2 \Big), \tag{2}$$

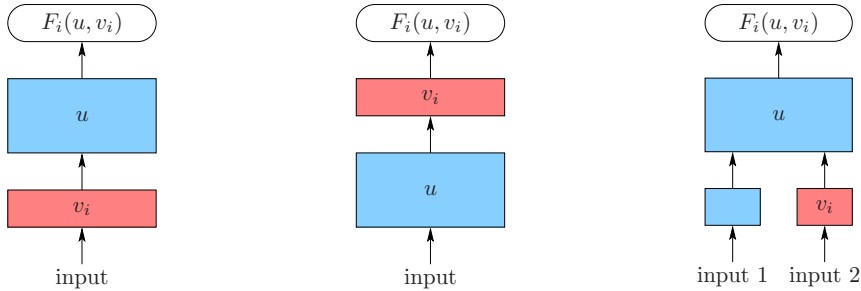

(a) Personalized input layer(s).     (b) Personalized output layer(s).     (c) Personalized split input layer(s).

Figure 1: Three simple examples of partitioning deep learning models.

where $w_i$ for $i = 1, \ldots, n$ are personalized model parameters at the devices, $w_0$ is a reference model maintained by the server, and the $\lambda_i$'s are regularization weights that control the extent of personalization. A major disadvantage of the formulation (2), which we call *full model personalization*, is that it requires twice the memory footprint of the model, $w_i$ and $w_0$ at each device, which severely limits the size of trainable models. On the other hand, the flexibility of full model personalization can be unnecessary. Modern deep learning models are composed of many simple functional units and are typically organized into layers or a more general interconnected architecture. Personalizing the "right" components, selected with domain knowledge, may result in a substantial benefit with only a small increase in memory footprint. In addition, partial model personalization can be less susceptible to "catastrophic forgetting" (McCloskey & Cohen, 1989), where a large model finetuned on a small local dataset forgets the original (non-personalized) task, leading to a degradation of test performance.

We propose a framework for FL with *partial model personalization*. Specifically, we partition the model parameters into two groups: the *shared* parameters $u \in \mathbb{R}^{d_0}$ and the *personal* parameters $v_i \in \mathbb{R}^{d_i}$ for $i = 1, \ldots, n$. The full model on device $i$ is denoted as $w_i = (u, v_i)$, and the local loss function is $F_i(u, v_i) = (1/N_i) \sum_{i=1}^{N_i} f_i\big((u, v_i), z_{i,j}\big)$. Our goal is to solve the optimization problem

$$\underset{u, \, \{v_i\}_{i=1}^n}{\text{minimize}} \quad \sum_{i=1}^{n} \alpha_i F_i(u, v_i). \tag{3}$$

Notice that the dimensions of $v_i$ can be different across the devices, allowing the personal components of the model to have different number of parameters or even different architecture.

We investigate two FL algorithms for solving problem (3): *FedSim*, a simultaneous update algorithm and *FedAlt*, an alternating update algorithm. Both algorithms follow the standard FL protocol. During each round, the server randomly selects a subset of the devices for update and broadcasts the current global version of the shared parameters to devices in the subset. Each selected device then performs one or more steps of (stochastic) gradient descent to update both the shared parameters and the personal parameters, and sends the updated shared parameters to the server for aggregation. The updated personal parameters are kept local at the device to serve as the initial states when the device is selected for another update. In FedSim, the shared and personal parameters are updated simultaneously during each local iteration. In FedAlt, the devices first update the personal parameters with the received shared parameters fixed and then update the shared parameters with the new personal parameters fixed. We provide convergence analysis and empirical evaluation of both methods.

The main contributions of this paper are summarized as follows:

- We propose a general framework of *FL with partial model personalization*, which relies on domain knowledge to select a small portion of the model to personalize, thus imposing a much smaller memory footprint on the devices than full model personalization. This framework unifies existing work on personalized FL and allows arbitrary partitioning of deep learning models.
- We provide *convergence guarantees* for the FedSim and FedAlt methods in the general (smooth) nonconvex setting. While both methods have appeared in the literature previously, they are either used without convergence analysis or with results on limited settings (assuming convexity or full participation) Our analysis provides theoretical support for the general nonconvex setting with partial participation. The analysis of FedAlt with partial participation is especially challenging and we develop a novel technique of *virtual full participation* to overcome the difficulties.

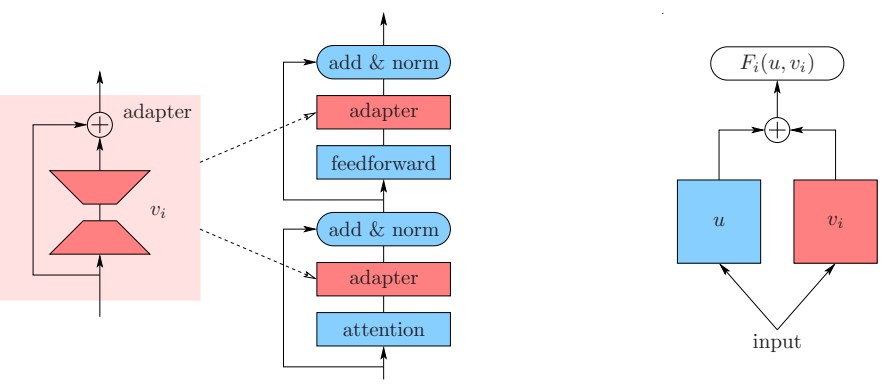

(a) Transformer layer with two adapters.
(b) Generalized additive model.

Figure 2: More structured partial model personalization. (a) The adapter has a skip connection, thus it collapses to the identity mapping if $v_i = 0$; in addition, it has a bottleneck in the middle (Houlsby et al., 2019). (b) The generalized additive model can be further augmented with a shared input layer for representation learning.

- We conduct *extensive experiments* on image classification and text prediction tasks, exploring different model personalization strategies for each task, and comparing with several strong baselines. Our results demonstrate that partial model personalization can obtain most of the benefit of full model personalization with a small fraction of personalized parameters, and FedAlt often outperforms FedSim.
- Our experiments also reveal that personalization (full or partial) may lead to *worse performance for some devices*, despite improving the average. Typical forms of regularization such as weight decay and dropout do not mitigate this issue. This phenomenon has been overlooked in previous work and calls for future research to improve both performance and fairness.

**Related work.** Specific forms of partial model personalization have been considered in previous works. Liang et al. (2019) propose to personalize the input layers to learn a personalized representation per-device (Figure 1a), while Arivazhagan et al. (2019) and Collins et al. (2021) propose to personalize the output layer while learning a shared representation with the input layers (Figure 1b). Both FedSim and FedAlt have appeared in the literature before, but the scope of their convergence analysis is limited. Specifically, Liang et al. (2019), Arivazhagan et al. (2019) and Hanzely et al. (2021) use FedSim, while Collins et al. (2021) and Singhal et al. (2021) proposed variants of FedAlt. Notably, Hanzely et al. (2021) establish convergence of FedSim with *full* device participation in the convex and non-convex cases, while Collins et al. (2021) prove the linear convergence of FedAlt for a two-layer linear network where $F_i(\cdot, v_i)$ and $F_i(u, \cdot)$ are both convex for fixed $v_i$ and $u$ respectively. We analyze both FedSim and FedAlt in the general nonconvex case with partial device participation, hence addressing a more general and practical setting.

While we primarily consider the problem (3) in the context of partial model personalization, it can serve as a general formulation that covers many other problems. Hanzely et al. (2021) demonstrate that various full model personalization formulations based on regularization (Dinh et al., 2020; Li et al., 2021), including (2), as well as interpolation (Deng et al., 2020; Mansour et al., 2020) are special cases of this problem. The rates of convergence we prove in §3 are competitive with or better than those in previous works for full model personalization methods in the non-convex case.

## 2  PARTIALLY PERSONALIZED MODELS

Modern deep learning models all have a multi-layer architecture. While a complete understanding of why they work so well is still out of reach, a general insight is that the lower layers (close to the input) are mostly responsible for feature extraction and the upper layers (close to the output) focus on complex pattern recognition. Depending on the application scenarios and domain knowledge, we may personalize either the input layer(s) or the output layer(s) of the model; see Figure 1.

In Figure 1c, the input layers are split horizontally into two parts, one shared and the other personal. They process different chunks of the input vector and their outputs are concatenated before feeding

---

**Algorithm 1** Federated Learning with Partial Model Personalization (FedSim / FedAlt)

---

**Input:** initial states $u^{(0)}, \{v_i^{(0)}\}_{i=1}^n$, number of rounds $T$, number of devices per round $m$
1: **for** $t = 0, 1, \cdots, T - 1$ **do**
2:      server randomly samples $m$ devices as $S^{(t)} \subset \{1, \ldots, n\}$
3:      server broadcasts $u^{(t)}$ to each device in $S^{(t)}$
4:      **for** each device $i \in S^{(t)}$ in parallel, **do**
5:          $\left( u_i^{(t+1)}, v_i^{(t+1)} \right) = \text{LocalSim} / \text{LocalAlt}\left( u^{(t)}, v_i^{(t)} \right)$        $\triangleright \, v_i^{(t+1)} = v_i^{(t)}$ if $i \notin S^{(t)}$
6:          send $u_i^{(t+1)}$ back to server
7:      server updates $u^{(t+1)} = \frac{1}{m} \sum_{i \in S^{(t)}} u_i^{(t+1)}$

---

to the upper layers of the model. As demonstrated in (Bui et al., 2019), this partitioning can help protect user-specific private features (input 2 in Figure 1c) as the corresponding feature embedding (through $v_i$) are personalized and kept local at the device. Similar architectures have also been proposed in context-dependent language models (e.g., Mikolov & Zweig, 2012).

A more structured partitioning is illustrated in Figure 2a, where a typical transformer layer (Vaswani et al., 2017) is augmented with two adapters. This architecture is proposed by Houlsby et al. (2019) for finetuning large language models. Similar residual adapter modules are proposed by Rebuffi et al. (2017) for image classification models in the context of multi-task learning. In the context of FL, we treat the adapter parameters as personal and the rest of the model parameters as shared.

Figure 2b shows a generalized additive model, where the outputs of two separate models, one shared and the other personalized, are fused to generate a prediction. Suppose the shared model is $h(u, \cdot)$ and the personal model is $h_i(v_i, \cdot)$. For regression tasks with samples $z_{i,j} = (x_{i,j}, y_{i,j})$, where $x_{i,j}$ is the input and $y_{i,j} \in \mathbb{R}^p$ is the output, we let $F_i(u, v_i) = (1/N_i) \sum_{j=1}^{N_i} f_i\big((u, v_i), z_{i,j}\big)$ with

$$f_i\big((u, v_i), z_{i,j}\big) = \|y_{i,j} - h(u, x_{i,j}) - h_i(v_i, x_{i,j})\|^2.$$

In this special case, the personal model fits the residual of the shared model and vice-versa (Agarwal et al., 2020). For classification tasks, $h(u, \cdot)$ and $h_i(v_i, \cdot)$ produce probability distributions over multiple classes. We can use the cross-entropy loss between $y_{i,j}$ and a convex combination of the two model outputs: $\theta h(u, x_{i,j}) + (1 - \theta) h_i(v_i, x_{i,j})$, where $\theta \in (0, 1)$ is a learnable parameter.

Finally, we can cast the formulation (2) of full model personalization as a special case of (3) by letting

$$u \leftarrow w_0, \qquad v_i \leftarrow w_i, \qquad F_i(u, v_i) \leftarrow F_i(v_i) + (\lambda_i/2) \|v_i - u\|^2.$$

Many other formulations of full personalization can be reduced to (3); see Hanzely et al. (2021).

## 3    ALGORITHMS AND CONVERGENCE ANALYSIS

In this section, we present and analyze two FL algorithms for solving problem (3). To simplify presentation, we denote $V = (v_1, \ldots, v_n) \in \mathbb{R}^{d_1 + \cdots + d_n}$ and focus on the case of $\alpha_i = 1/n$, i.e.,

$$\text{minimize}_{u, V} \quad F(u, V) := \frac{1}{n} \sum_{i=1}^n F_i(u, v_i). \tag{4}$$

This is equivalent to (3) if we scale $F_i$ by $n\alpha_i$, thus does not lose generality. Moreover, we consider more general local functions $F_i(u, v_i) = \mathbf{E}_{z \sim \mathcal{D}_i}[f_i((u, v_i), z)]$, where $\mathcal{D}_i$ is the local distribution.

The FedSim and FedAlt algorithms share a common outer-loop description given in Algorithm 1. They differ only in the local update procedures LocalSim and LocalAlt, which are given in Algorithm 2 and Algorithm 3 respectively. In the two local update procedures, $\widetilde{\nabla}_u$ and $\widetilde{\nabla}_v$ represent stochastic gradients with respect to $w$ and $v_i$ respectively. In LocalSim (Algorithm 2), the personal variables $v_i$ and local version of the shared parameters $u_i$ are updated simultaneously, with their (stochastic) partial gradients evaluated at the same point. In LocalAlt (Algorithm 3), the personal parameters are updated first with the received shared parameters fixed, then the shared parameters are updated with the new personal parameters fixed. They are analogous to the classical Jacobi update and Gauss-Seidel update in numerical linear algebra (e.g., Demmel, 1997, §6.5).

In order to analyze the convergence of the two algorithms, we make the following assumptions.

---

**Algorithm 2** LocalSim$(u, v_i)$

---

**Input:** number of steps $\tau$, step sizes $\gamma_v$ and $\gamma_u$
 1: initialize $v_{i,0} = v_i$
 2: initialize $u_{i,0} = u$
 3: **for** $k = 0, 1, \cdots, \tau - 1$ **do**
 4:     $v_{i,k+1} = v_{i,k} - \gamma_v \widetilde{\nabla}_v F_i(u_{i,k}, v_{i,k})$
 5:     $u_{i,k+1} = u_{i,k} - \gamma_u \widetilde{\nabla}_u F_i(u_{i,k}, v_{i,k})$
 6: update $v_i^+ = v_{i,\tau}$
 7: update $u_i^+ = u_{i,\tau}$
 8: **return** $(u_i^+, v_i^+)$

---

**Algorithm 3** LocalAlt$(u, v_i)$

---

**Input:** number of steps $\tau_v, \tau_u$, step sizes $\gamma_v, \gamma_u$
 1: initialize $v_{i,0} = v_i$
 2: **for** $k = 0, 1, \cdots, \tau_v - 1$ **do**
 3:     $v_{i,k+1} = v_{i,k} - \gamma_v \widetilde{\nabla}_v F_i(u, v_{i,k})$
 4: update $v_i^+ = v_{i,\tau_v}$ and initialize $u_{i,0} = u$
 5: **for** $k = 0, 1, \cdots, \tau_u - 1$ **do**
 6:     $u_{i,k+1} = u_{i,k} - \gamma_u \widetilde{\nabla}_u F_i(u_{i,k}, v_i^+)$
 7: update $u_i^+ = u_{i,\tau_u}$
 8: **return** $(u_i^+, v_i^+)$

---

**Assumption 1** (Smoothness). *The function $F_i$ is continuously differentiable for each $i = 1, \ldots, n$, and there exist constants $L_u$, $L_v$, $L_{uv}$ and $L_{vu}$ such that for each $i = 1, \ldots, n$, it holds that*

- *$\nabla_u F_i(u, v_i)$ is $L_u$-Lipschitz with respect to $u$ and $L_{uv}$-Lipschitz with respect to $v_i$;*

- *$\nabla_v F_i(u, v_i)$ is $L_v$-Lipschitz with respect to $v_i$ and $L_{vu}$-Lipschitz with respect to $u$.*

Due to the definition of $F(u, V)$ in (4), it is easy to verify that $\nabla_u F(u, V)$ has Lipschitz constant $L_u$ with respect to $u$, $L_{uv}/\sqrt{n}$ with respect to $V$, and $L_{uv}/n$ with respect to any $v_i$. We also define

$$\chi := \max\{L_{uv}, L_{vu}\}/\sqrt{L_u L_v}, \tag{5}$$

which measures the relative cross-sensitivity of $\nabla_u F_i$ with respect to $v_i$ and $\nabla_v F_i$ with respect to $u$.

**Assumption 2** (Bounded Variance). *The stochastic gradients in Algorithm 2 and Algorithm 3 are unbiased and have bounded variance. That is, for all $u$ and $v_i$,*

$$\mathbf{E}\big[\widetilde{\nabla}_u F_i(u, v_i)\big] = \nabla_u F_i(u, v_i), \qquad \mathbf{E}\big[\widetilde{\nabla}_v F_i(u, v_i)\big] = \nabla_v F_i(u, v_i).$$

*Furthermore, there exist constants $\sigma_u$ and $\sigma_v$ such that*

$$\mathbf{E}\big[\big\|\widetilde{\nabla}_u F_i(u, v_i) - \nabla_u F_i(u, v_i)\big\|^2\big] \le \sigma_u^2, \qquad \mathbf{E}\big[\big\|\widetilde{\nabla}_v F_i(u, v_i) - \nabla_v F_i(u, v_i)\big\|^2\big] \le \sigma_v^2.$$

We can view $\nabla_u F_i(u, v_i)$, when $i$ is randomly sampled from $\{1, \ldots, n\}$, as a stochastic partial gradient of $F(u, V)$ with respect to $u$. The following assumption imposes a variance bound.

**Assumption 3** (Partial Gradient Diversity). *There exist $\delta \ge 0$ and $\rho \ge 0$ such that for all $u$ and $V$,*

$$\frac{1}{n} \sum_{i=1}^n \big\|\nabla_u F_i(u, v_i) - \nabla_u F(u, V)\big\|^2 \le \delta^2 + \rho^2 \big\|\nabla_u F(u, V)\big\|^2.$$

With $\rho = 0$, this assumption is similar to a constant variance bound on the stochastic gradient $\nabla_u F_i(u, v_i)$; with $\rho > 0$, it allows the variance to grow with the norm of the full gradient.

Throughout this paper, we assume $F$ is bounded below by $F^\star$ and denote $\Delta F_0 = F(u^{(0)}, V^{(0)}) - F^\star$. Further, we use shorthand $V^{(t)} = (v_1^{(t)}, \ldots, v_n^{(t)})$ and

$$\Delta_u^{(t)} = \big\|\nabla_u F(u^{(t)}, V^{(t)})\big\|^2, \quad \text{and} \quad \Delta_v^{(t)} = \frac{1}{n} \sum_{i=1}^n \big\|\nabla_v F_i(u^{(t)}, v_i^{(t)})\big\|^2.$$

For smooth and nonconvex loss functions $F_i$, we obtain convergence in expectation to a stationary point of $F$ if the expected values of these two sequences converge to zero.

We first present our main result for FedSim (Algorithm 1 with LocalSim), proved in Appendix A.2.

**Theorem 1** (**Convergence of FedSim**). *Suppose Assumptions 1, 2 and 3 hold and the learning rates in FedSim are chosen as $\gamma_u = \eta/(L_u \tau)$ and $\gamma_v = \eta/(L_v \tau)$ with*

$$\eta \le \min\left\{ \frac{1}{12(1+\chi^2)(1+\rho^2)}, \ \sqrt{\frac{m/n}{196(1-\tau^{-1})(1+\chi^2)(1+\rho^2)}} \right\}.$$

*Then, ignoring absolute constants, we have*

$$\frac{1}{T} \sum_{t=0}^{T-1} \left( \frac{1}{L_u} \mathbf{E}\big[\Delta_u^{(t)}\big] + \frac{m}{nL_v} \mathbf{E}\big[\Delta_v^{(t)}\big] \right) \le \frac{\Delta F_0}{\eta T} + \eta(1+\chi^2)\left( \frac{\sigma_u^2 + \delta^2(1 - \frac{m}{n})}{mL_u} + \frac{m\sigma_v^2}{nL_v} \right)$$

$$+ \eta^2(1 - \tau^{-1})(1+\chi^2)\left( \frac{\sigma_u^2 + \delta^2}{L_u} + \frac{\sigma_v^2}{L_v} \right). \tag{6}$$

Table 1: Convergence rates of FedSim and FedAlt in different regimes along with optimal learning rate via $\eta$. We only show the dominant terms and hide the lower order terms in $T$ for simplicity.

| Condition | $\eta$ | FedSim | FedAlt |
|---|---|---|---|
| General | $T^{-1/2}$ | $\frac{\Delta F_0}{\sqrt{T}} + \frac{1+\chi^2}{\sqrt{T}}\left(\frac{\sigma_u^2 + \delta^2\left(1-\frac{m}{n}\right)}{mL_u} + \frac{m\sigma_v^2}{nL_v}\right)$ | $\frac{\Delta F_0}{\sqrt{T}} + \frac{\sigma_u^2 + \delta^2\left(1-\frac{m}{n}\right)}{mL_u\sqrt{T}} + \frac{\sigma_v^2}{L_v\sqrt{T}}\frac{m+\chi^2(n-m)}{n}$ |
| $\sigma_u^2 = \sigma_v^2 = 0$ $m=n,\ \tau>1$ | $T^{-1/3}$ | $\frac{1}{T^{2/3}}\left(\Delta F_0 + \frac{\delta^2(1-\tau^{-1})(1+\chi^2)}{L_u}\right)$ | $\frac{1}{T^{2/3}}\left(\Delta F_0 + \frac{\delta^2(1-\tau^{-1})}{L_u}\right)$ |
| $\sigma_u^2 = \sigma_v^2 = 0$ $m=n,\ \tau=1$ | $O(1)$ | $\frac{\Delta F_0}{T}$ | $\frac{\Delta F_0}{T}$ |

The left-hand side of (6) is the average over time of a weighted sum of $\mathbf{E}\big[\Delta_u^{(t)}\big]$ and $\mathbf{E}\big[\Delta_v^{(t)}\big]$. The right-hand side contains three terms of order $O(1/(\eta T))$, $O(\eta)$ and $O(\eta^2)$ respectively. We can minimize the right-hand side by optimizing over $\eta$. By considering special cases such as $\sigma_u^2 = \sigma_v^2 = 0$ and $m = n$, some terms on the right-hand side disappear and we can obtain improved rates. Table 1 shows the results in several different regimes along with the optimal choices of $\eta$.

**Challenge in Analyzing FedAlt.** We now turn to FedAlt. Note that the personal parameters are updated only for the $m$ selected devices in $S^{(t)}$ in each round $t$. Specifically,

$$v_i^{(t+1)} = \begin{cases} v_i^{(t)} - \gamma_v \sum_{k=0}^{\tau_v} \widetilde{\nabla}_v F_i\big(u^{(t)}, v_{i,k}^{(t)}\big) & \text{if } i \in S^{(t)}, \\ v_i^{(t)} & \text{if } i \notin S^{(t)}. \end{cases}$$

Consequently, the vector $V^{(t+1)}$ of personal parameters depends on the random variable $S^{(t)}$. This makes it challenging to analyze the $u$-update steps of FedAlt because they are performed after $V^{(t+1)}$ is generated (as opposed to simultaneously in FedSim). When we take expectations with respect to the sampling of $S^{(t)}$ in analyzing the $u$-updates, $V^{(t+1)}$ becomes a dependent random variable, which prevents standard proof techniques from going through (see details in Appendix A.3).

We develop a novel technique called *virtual full participation* to overcome this challenge. Specifically, we define a virtual vector $\widetilde{V}^{(t+1)}$, which is the result if *every* device were to perform local $v$-updates. It is independent of the sampling of $S^{(t)}$ and we can derive a convergence rate for related quantities. We carefully translate this rate from the virtual $\widetilde{V}^{(t+1)}$ to the actual $V^{(t)}$ to get the following result.

**Theorem 2** (**Convergence of FedAlt**). *Suppose Assumptions 1, 2 and 3 hold and the learning rates in FedAlt are chosen as $\gamma_u = \eta/(L_u\tau_u)$ and $\gamma_v = \eta/(L_v\tau_v)$, with*

$$\eta \le \min\left\{\frac{1}{24(1+\rho^2)}, \frac{m}{128\chi^2(n-m)}, \sqrt{\frac{m}{\chi^2 n}}\right\}.$$

*Then, ignoring absolute constants, we have*

$$\frac{1}{T}\sum_{t=0}^{T-1}\left(\frac{1}{L_u}\mathbf{E}\big[\Delta_u^{(t)}\big] + \frac{m}{nL_v}\mathbf{E}\big[\Delta_v^{(t)}\big]\right) \le \frac{\Delta F_0}{\eta T} + \eta\left(\frac{\sigma_u^2 + \delta^2\left(1-\frac{m}{n}\right)}{mL_u} + \frac{\sigma_v^2}{L_v}\frac{m+\chi^2(n-m)}{n}\right)$$

$$+ \eta^2\left(\frac{\sigma_u^2+\delta^2}{L_u}(1-\tau_u^{-1}) + \frac{\sigma_v^2 m}{L_v n}(1-\tau_v^{-1}) + \frac{\chi^2\sigma_v^2}{L_v}\right).$$

The proof of Theorem 2 is given in Appendix A.3. Similar to the results for FedSim, we can choose $\eta$ to minimize the above upper bound to obtain the best convergence rate, as summarized in Table 1.

**Comparing FedSim and FedAlt.** Table 1 shows that both FedSim and FedAlt exhibit the standard $O(1/\sqrt{T})$ rate in the general case. Comparing the constants in their rates, we identify two regimes in terms of problem parameters. The regime where FedAlt dominates FedSim is characterized by

$$\frac{\sigma_v^2}{L_v}\left(1 - \frac{2m}{n}\right) < \frac{\sigma_u^2 + \delta^2(1-m/n)}{mL_u}.$$

A practically relevant scenario where this is true is $\sigma_v^2 \approx 0$ and $\sigma_u^2 \approx 0$ from using large or full batch on a small number of samples per device. Here, the rate of FedAlt is better than FedSim by a factor of $(1+\chi^2)$, indicating that the rate of FedAlt is less affected by the coupling between the personal and shared parameters. Our experiments in §4 corroborate the practical relevance of this regime.

The rates from Table 1 also apply for full personalization schemes without convergence guarantees in the nonconvex case (Agarwal et al., 2020; Mansour et al., 2020; Li et al., 2021). Our rates are better than those of (Dinh et al., 2020) for their pFedMe objective.

Table 2: Summary of datasets and models. A histogram of data per device is given in Figure 5 (Appendix B).

| Task | Dataset | #Classes | Model | # Model Params | #Devices | #Data per device Mean | Max |
|------|---------|----------|-------|----------------|----------|------|-----|
| Next-word prediction | StackOverflow | 10000 | 4-layer transformer | $6M$ | 1000 | 4964 | 15520 |
| Landmark recognition | GLDv2 | 2028 | ResNet-18 | $12M$ | 823 | 88 | 1000 |
| Character recognition | EMNIST | 63 | ResNet-18 | $11M$ | 1114 | 298 | 418 |

## 4 EXPERIMENTS

In this section, we experimentally compare different model personalization schemes using FedAlt and FedSim as well as no model personalization. Details about the experiments, hyperparameters and additional results are provided in the appendices. The code to reproduce the experimental results will be publicly released.

**Datasets, Tasks and Models.** We consider three learning tasks; they are summarized in Table 2.

(a) *Next-Word Prediction*: We use the StackOverflow dataset, where each device corresponds to the questions and answers of one user on `stackoverflow.com`. This task is representative of mobile keyboard predictions. We use a 4-layer transformer model (Vaswani et al., 2017).

(b) *Visual Landmark Recognition*: We use the GLDv2 dataset (Weyand et al., 2020; Hsu et al., 2020), a large-scale dataset with real images of global landmarks. Each device corresponds to a Wikipedia contributor who uploaded images. This task resembles a scenario where smartphone users capture images of landmarks while traveling. We use a ResNet-18 (He et al., 2016) model with group norm instead of batch norm (Hsieh et al., 2020) and images are reshaped to $224 \times 224$.

(c) *Character Recognition*: We use the EMNIST dataset (Cohen et al., 2017), where the input is a $28 \times 28$ grayscale image of a handwritten character and the output is its label (0-9, a-z, A-Z). Each device corresponds to a writer of the character. We use a ResNet-18 model, with input and output layers modified to accommodate the smaller image size and number of classes.

All models are trained with the cross entropy loss and evaluated with top-1 accuracy of classification.

**Model Partitioning for Partial Personalization.** We consider three partitioning schemes.

(a) *Input layer personalization*: This architecture learns a personalized representation per-device by personalizing the input layer, while the rest of the model is shared (Figure 1a). For the transformer, we use the first transformer layer in place of the embedding layer.

(b) *Output layer personalization*: This architecture learns a shared representation but personalizes the prediction layer (Figure 1b). For the transformer model, we use the last transformer layer instead of the output layer.

(c) *Adapter personalization*: In this architecture, each device adds lightweight personalized adapter modules between specific layers of a shared model (Figure 2a). We use the transformer adapters of Houlsby et al. (2019) and for ResNet-18, the residual adapters of Rebuffi et al. (2017).

**Algorithms and Experimental Pipeline.** For full model personalization, we consider three baselines: (i) *Finetune*, where each device finetunes (using SGD locally) its personal full model starting from a learned common model, (ii) *Ditto* (Li et al., 2021), which is finetuning with $\ell_2$ regularization, and, (iii) *pFedMe* (Dinh et al., 2020) which minimizes the objective (2). All methods, including FedSim, FedAlt and the baselines are initialized with a global model trained with FedAvg.

### 4.1 EXPERIMENTAL RESULTS

**Partial personalization nearly matches full personalization and can sometimes outperform it.** Table 3 shows the *average* test accuracy across all devices of different FL algorithms. We see that on the StackOverflow dataset, output layer personalization (25.05%) makes up nearly 90% of the gap between the non-personalized baseline (23.82%) and full personalization (25.21%). On EMNIST, adapter personalization exactly matches full personalization. Most surprisingly, on GLDv2, adapter personalization outperforms full personalization by 3.5pp (percentage points).

Table 3: Comparison of partial model personalization with full model personalization in terms of the *average* test accuracy % across all devices. The subscript denotes the standard deviation over 5 runs with different random seeds. The boldfaced/highlighted numbers denote the accuracies within one standard deviation of the maximum in each row. For partial personalization, we show the accuracy of FedAlt; see Table 4 for FedSim.

| | Non-pers. | Full Model Personalization | | | Partial Model Personalization | | |
|---|---|---|---|---|---|---|---|
| | FedAvg | Finetune | Ditto | pFedMe | Input Layer | Output Layer | Adapter |
| StackOverflow | 23.82 | $\mathbf{25.20}_{0.01}$ | $\mathbf{25.20}_{0.01}$ | $\mathbf{25.21}_{0.01}$ | $24.44_{0.01}$ | $25.05_{0.01}$ | $24.82_{0.01}$ |
| GLDv2 | 51.43 | $62.85_{0.02}$ | $62.85_{0.01}$ | $62.92_{0.02}$ | $53.94_{0.07}$ | $56.64_{0.05}$ | $\mathbf{66.41}_{0.06}$ |
| EMNIST | 93.18 | $\mathbf{94.13}_{0.01}$ | $\mathbf{94.13}_{0.01}$ | $\mathbf{94.13}_{0.01}$ | $93.62_{0.04}$ | $93.57_{0.05}$ | $\mathbf{94.13}_{0.03}$ |

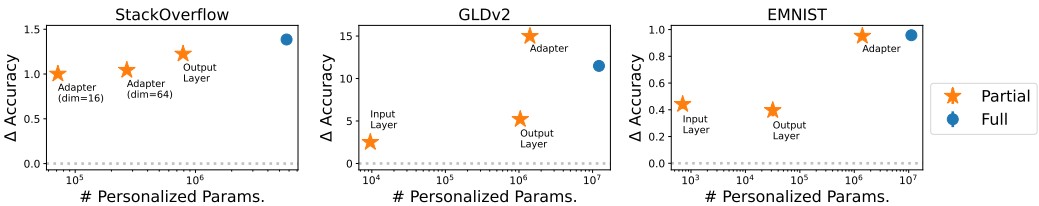

Figure 3: Absolute change in accuracy (percentage points) due to personalization plotted against number of personal parameters (i.e., dimensionality of $v_i$). Note that the $x$-axis is in log scale.

This success of adapter personalization can be explained partly by the nature of GLDv2. On average, the training data on each device contains 25 classes out of a possible 2028 while the testing data contains 10 classes not seen in its own training data. These unseen classes account for nearly 23% of all testing data. Personalizing the full model is susceptible to "forgetting" the original task (Kirkpatrick et al., 2017), making it harder to get these unseen classes right. Such *catastrophic forgetting* is worse when finetuning on a very small local dataset, as we often have in FL. On the other hand, personalizing the adapters does not suffer as much from this issue (Rebuffi et al., 2017).

**Partial personalization only requires a fraction of the parameters to be personalized.** Figure 3 shows that the number of personalized parameters required to compete with full model personalization is rather small. On StackOverflow, personalizing 1.2% of the parameters with adapters captures 72% of the accuracy boost from personalizing all $5.7M$ parameters; this can be improved to nearly 90% by personalizing 14% of the parameters (output layer). Likewise, we match full personalization on EMNIST and exceed it on GLDv2 with adapters, personalizing 11.5-12.5% of parameters.

**The best personalized architecture is model and task dependent.** Table 3 shows that personalizing the final transformer layer (denoted as "Output Layer") achieves the best performance for StackOverflow, while the residual adapter achieves the best performance for GLDv2 and EMNIST. This shows that the approach of personalizing a fixed model part, as in several past works, is suboptimal. Our framework allows for the use of domain knowledge to determine customized personalization.

**Finetuning is competitive with other full personalization methods.** Full finetuning matches the performance of pFedMe and Ditto on StackOverflow and EMNIST. On GLDv2, however, pFedMe outperforms finetuning by 0.07pp, but is still 3.5pp worse than adapter personalization.

**FedAlt outperforms FedSim for partial personalization.** If the optimization problem (3) were convex, we would expect similar performance from FedAlt and FedSim. However, with nonconvex optimization problems such as the ones considered here, the choice of the optimization algorithm often affects the quality of the solution found. We see from Table 4 that FedAlt is almost always better than FedSim by a small margin, e.g., 0.08pp for StackOverflow/Adapter and 0.3pp for GLDv2/Input Layer. FedSim in turn yields a higher accuracy than simply finetuning the personalized part of the model, by a large margin, e.g., 0.12pp for StackOverflow/Output Layer and 2.55pp for GLDv2/Adapter.

Table 4: Comparing FedAlt and FedSim for partial model personalization. "FT (part.)" means finetuning the personal parameters $v_i$ while fixing the shared parameters $u$ from non-personalized training. The numbers are averaged over 5 runs with different random seeds. The standard deviations are given in Table 9 (Appendix C).

|  | StackOverflow | | | GLDv2 | | | EMNIST | | |
|---|---|---|---|---|---|---|---|---|---|
|  | FT (part.) | FedAlt | FedSim | FT (part.) | FedAlt | FedSim | FT (part.) | FedAlt | FedSim |
| Input Layer | **24.96** | 24.44 | 24.81 | 51.97 | **53.94** | 53.64 | 93.29 | **93.62** | 93.55 |
| Output Layer | 24.93 | **25.05** | 25.02 | 53.21 | **56.64** | 56.24 | 93.37 | **93.57** | 93.55 |
| Adapter | 24.71 | **24.82** | 24.74 | 63.86 | **66.41** | 66.35 | 93.66 | **94.13** | 94.07 |

## 4.2 Effects of personalization on per-device generalization

**Personalization hurts the test accuracy on some devices.** Figure 4 shows the change in training and test accuracy of each device, compared with a non-personalized model trained by FedAvg. We see that personalization leads to an improvement in training accuracy across all devices, but *a reduction in test accuracy* on some of the devices over the non-personalized baseline. In particular, devices whose testing performance is hurt by personalization are mostly on the left side of the plot, meaning that they have relatively small number of training samples. On the other hand, many devices with the most improved test accuracy also appear on the left side, signaling the benefit of personalization. Therefore, there is a large variation of results for devices with few samples.

Additional experiments (see Appendix C) show that using $\ell_2$ regularization, as in (2), or weight decay does not mitigate this issue. In particular, increasing regularization strength (less personalization) can reduce the spread of per-device accuracy, but only leads to a worse average accuracy that is close to using a common model. Other simple strategies such as dropout also do not fix this issue.

An ideal personalized method would boost performance on most of the devices without causing a reduction in (test) accuracy on any device. Realizing this goal calls for a sound statistical analysis for personalized FL and may require sophisticated methods for local performance diagnosis and more structured regularization. These are very promising directions for future research.

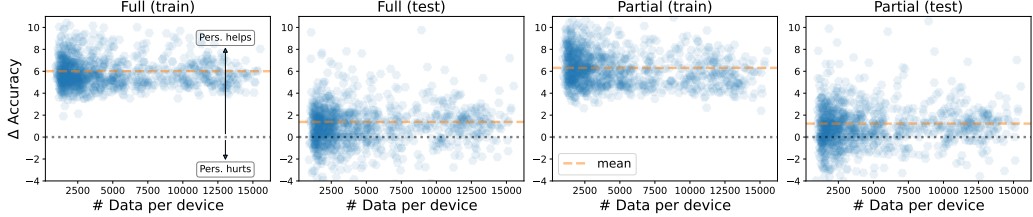

Figure 4: StackOverflow task: Scatter plot of change in training and test accuracy (percentage points) per-device versus the number of training samples on the device for (a) **Left**: full personalization with finetuning, and, (b) **Right**: partial personalization with the output layer.

## 5 Discussion

In addition to a much smaller memory footprint than full model personalization and being less susceptible to catastrophic forgetting, partial model personalization has other advantages. For example, it reduces the amount communication between the server and the devices because only the shared parameters are transmitted. While the communication saving may not be significant (especially when the personal parameters are only a small fraction of the full model), communicating only the shared parameters may have significant implications for privacy. Intuitively, it can be harder to infer private information from partial model information. This is especially the case if the more sensitive features of the data are processed through personal components of the model that are kept local at the devices. For example, we speculate that less noise needs to be added to the communicated parameters in order to satisfy differential privacy requirements (Abadi et al., 2016). This is a very promising direction for future research.

REPRODUCIBILITY STATEMENT

For theoretical results, we state and discuss the assumptions in Appendix A. The full proofs of all theoretical statements are also given there.

For our numerical results, we take multiple steps for reproducibility. First, we run each numerical experiment for five random seeds, and report both the mean and standard deviation over these runs. Second, we only use publicly available datasets and report the preprocessing at length in Appendix B. Third, we give the full list of hyperparameters used in our experiments in Table 8 in Appendix B. Finally, we will publicly release the code to reproduce the our experimental results.

ETHICS STATEMENT

The proposed framework for partial model personalization is immediately applicable for a range of practical federated learning applications in edge devices such as text prediction and speech recognition. One of key considerations of federated learning is privacy. Partial model personalization maintains the all the privacy benefits of current non-personalized federated learning systems. Indeed, our approach is compatible with techniques to enhance privacy such as differential privacy and secure aggregation. We also speculate that partial personalization has the potential for further reducing the privacy footprint — an investigation of this subject is beyond the scope of this work and is an interesting direction for future work.

On the flip side, we also observed in experiments that personalization (both full or partial) leads to a reduction in test performance on some of the devices. This has important implications for fairness, and calls for further research into the statistical aspects of personalization, performance diagnostics as well as more nuanced definitions of fairness in federated learning.

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

# Appendix

## Table of Contents

# A CONVERGENCE ANALYSIS: FULL PROOFS

We give the full convergence proofs here. The outline of this section is:

- §A.1: Review of setup and assumptions;
- §A.2: Convergence analysis of FedSim and the full proof of Theorem 1;
- §A.3: Convergence analysis of FedAlt and the full proof of Theorem 2;
- §A.4: Technical lemmas used in the analysis.

## A.1 REVIEW OF SETUP AND ASSUMPTIONS

We consider a federated learning system with $n$ devices. Let the loss function on device $i$ be $F_i(u, v_i)$, where $u \in \mathbb{R}^{d_0}$ denotes the shared parameters across all devices and $v_i \in \mathbb{R}^{d_i}$ denotes the personal parameters at device $i$. We aim to minimize the function

$$F(u, V) := \frac{1}{n} \sum_{i=1}^{n} F_i(u, v_i), \tag{7}$$

where $V = (v_1, \cdots, v_n)$ is a concatenation of all the personalized parameters. This is a special case of (3) with the equal per-device weights, i.e., $\alpha_i = 1/n$. Recall that we assume that $F$ is bounded from below by $F^\star$.

For convenience, we reiterate Assumptions 1, 2 and 3 from the main-paper as Assumptions $1'$, $2'$ and $3'$ below respectively, with some additional comments and discussion.

**Assumption $1'$** (Smoothness). *For each device $i = 1, \ldots, n$, the objective $F_i$ is smooth, i.e., it is continuously differentiable and,*

*(a) $u \mapsto \nabla_u F_i(u, v_i)$ is $L_u$-Lipschitz for all $v_i$,*
*(b) $v_i \mapsto \nabla_v F_i(u, v_i)$ is $L_v$-Lipschitz for all $u$,*
*(c) $v_i \mapsto \nabla_u F_i(u, v_i)$ is $L_{uv}$-Lipschitz for all $u$, and,*
*(d) $u \mapsto \nabla_v F_i(u, v_i)$ is $L_{vu}$-Lipschitz for all $v_i$.*

*Further, we assume for some $\chi > 0$ that*

$$\max\{L_{uv}, L_{vu}\} \le \chi \sqrt{L_u L_v}.$$

The smoothness assumption is a standard one. We can assume without loss of generality that the cross-Lipschitz coefficients $L_{uv}, L_{vu}$ are equal. Indeed, if $F_i$ is twice continuously differentiable, we can show that $L_{uv}, L_{vu}$ are both equal to the operator norm $\|\nabla^2_{uv} F_i(u, v_i)\|_{\mathrm{op}}$ of the mixed second derivative matrix. Further, $\chi$ denotes the extent to which $u$ impacts the gradient of $v_i$ and vice-versa.

Our next assumption is about the variance of the stochastic gradients, and is standard in literature. Compared to the main paper, we adopt a more precise notation about stochastic gradients.

**Assumption $2'$** (Bounded Variance). *Let $\mathcal{D}_i$ denote a probability distribution over the data space $\mathcal{Z}$ on device $i$. There exist functions $G_{i,u}$ and $G_{i,v}$ which are unbiased estimates of $\nabla_u F_i$ and $\nabla_v F_i$ respectively. That is, for all $u, v_i$:*

$$\mathbf{E}_{z \sim \mathcal{D}_i}[G_{i,u}(u, v, z)] = \nabla_u F_i(u, v_i), \quad \text{and} \quad \mathbf{E}_{z \sim \mathcal{D}_i}[G_{i,v}(u, v, z)] = \nabla_v F_i(u, v_i).$$

*Furthermore, the variance of these estimators is at most $\sigma_u^2$ and $\sigma_v^2$ respectively. That is,*

$$\mathbf{E}_{z \sim \mathcal{D}_i}\|G_{i,u}(u, v, z) - \nabla_u F_i(u, v_i)\|^2 \le \sigma_u^2,$$

$$\mathbf{E}_{z \sim \mathcal{D}_i}\|G_{i,v}(u, v, z) - \nabla_v F_i(u, v_i)\|^2 \le \sigma_v^2.$$

In practice, one usually has $G_{i,u}(u, v_i, z) = \nabla_u f_i((u, v_i), z)$, which is the gradient of the loss on datapoint $z \sim \mathcal{D}_i$ under the model $(u, v_i)$, and similarly for $G_{i,v}$.

Finally, we make a gradient diversity assumption.

**Assumption $3'$** (Partial Gradient Diversity). *There exist $\delta \ge 0$ and $\rho \ge 0$ such that for all $u$ and $V$,*

$$\frac{1}{n} \sum_{i=1}^{n} \|\nabla_u F_i(u, v_i) - \nabla_u F(u, V)\|^2 \le \delta^2 + \rho^2 \|\nabla_u F(u, V)\|^2. \tag{8}$$

---

**Algorithm 4** FedSim: Simultaneous update of shared and personal parameters

---

**Input:** Initial iterates $u^{(0)}, V^{(0)}$, Number of communication rounds $T$, Number of devices per round $m$, Number of local updates $\tau$, Local step sizes $\gamma_u, \gamma_v$.

1: **for** $t = 0, 1, \cdots, T-1$ **do**
2:      Sample $m$ devices from $[n]$ without replacement in $S^{(t)}$
3:      **for** each selected device $i \in S^{(t)}$ in parallel **do**
4:          Initialize $v_{i,0}^{(t)} = v_i^{(t)}$ and $u_{i,0}^{(t)} = u^{(t)}$
5:          **for** $k = 0, \cdots, \tau - 1$ **do**              ▷ Update all parameters jointly
6:              Sample data $z_{i,k}^{(t)} \sim \mathcal{D}_i$
7:              $v_{i,k+1}^{(t)} = v_{i,k}^{(t)} - \gamma_v G_{i,v}(u_{i,k}^{(t)}, v_{i,k}^{(t)}, z_{i,k}^{(t)})$
8:              $u_{i,k+1}^{(t)} = u_{i,k}^{(t)} - \gamma_u G_{i,u}(u_{i,k}^{(t)}, v_{i,k}^{(t)}, z_{i,k}^{(t)})$
9:          Update $v_i^{(t+1)} = v_{i,\tau}^{(t)}$ and $u_i^{(t+1)} = u_{i,\tau}^{(t)}$
10:      Update $u^{(t+1)} = \frac{\sum_{i \in S^{(t)}} \alpha_i u_i^{(t+1)}}{\sum_{i \in S^{(t)}} \alpha_i}$ at the server with secure aggregation
11: **return** $u^{(T)}, v_1^{(T)}, \cdots, v_n^{(T)}$

---

This assumption is analogous to the bounded variance assumption (Assumption 2′), but with the stochasticity coming from the sampling of devices. It characterizes how much local steps on one device help or hurt convergence globally. Similar gradient diversity assumptions are often used for analyzing non-personalized federated learning (Koloskova et al., 2020; Karimireddy et al., 2020). Finally, it suffices for the partial gradient diversity assumption to only hold at the iterates $(u^{(t)}, V^{(t)})$ generated by either FedSim or FedAlt.

A.2    CONVERGENCE ANALYSIS OF FEDSIM

We give the full form of FedSim in Algorithm 4 for the general case of unequal $\alpha_i$'s but focus on $\alpha_i = 1/n$ for the analysis. In order to simplify presentation, we denote $V^{(t)} = (v_1^{(t)}, \ldots, v_n^{(t)})$ and define the following shorthand for gradient terms

$$\Delta_u^{(t)} = \left\| \nabla_u F\left(u^{(t)}, V^{(t)}\right) \right\|^2, \quad \text{and} \quad \Delta_v^{(t)} = \frac{1}{n} \sum_{i=1}^n \left\| \nabla_v F_i\left(u^{(t)}, v_i^{(t)}\right) \right\|^2.$$

For convenience, we restate Theorem 1 from the main paper.

**Theorem 1** (**Convergence of FedSim**). *Suppose Assumptions 1, 2 and 3 hold and the learning rates in FedSim are chosen as $\gamma_u = \eta/(L_u \tau)$ and $\gamma_v = \eta/(L_v \tau)$ with*

$$\eta \le \min\left\{ \frac{1}{12(1+\chi^2)(1+\rho^2)}, \ \sqrt{\frac{m/n}{196(1-\tau^{-1})(1+\chi^2)(1+\rho^2)}} \right\}.$$

*Then, ignoring absolute constants, we have*

$$\frac{1}{T} \sum_{t=0}^{T-1} \left( \frac{1}{L_u} \mathbf{E}\big[\Delta_u^{(t)}\big] + \frac{m}{nL_v} \mathbf{E}\big[\Delta_v^{(t)}\big] \right) \le \frac{\Delta F_0}{\eta T} + \eta(1+\chi^2)\left( \frac{\sigma_u^2 + \delta^2\left(1 - \frac{m}{n}\right)}{mL_u} + \frac{m\sigma_v^2}{nL_v} \right)$$

$$+ \eta^2(1 - \tau^{-1})(1+\chi^2)\left( \frac{\sigma_u^2 + \delta^2}{L_u} + \frac{\sigma_v^2}{L_v} \right). \quad (6)$$

Before proving the theorem, we give the following corollary with optimized learning rates.

**Corollary 3.** *Consider the setting of Theorem 1 and let $\varepsilon > 0$ be given. Suppose we set the learning rates $\gamma_u = \eta/(\tau L_u)$ and $\gamma_v = \eta/(\tau L_v)$, where (ignoring absolute constants),*

$$\eta = \frac{\varepsilon}{\left( \frac{\delta^2}{L_u}\left(1 - \frac{m}{n}\right) + \frac{\sigma_u^2}{L_u} + \frac{\sigma_v^2 m}{L_u n} \right)(1+\chi^2)} \bigwedge \left( \frac{\varepsilon}{\left( \frac{\delta^2}{L_u} \vee \frac{\sigma_u^2}{L_u} \vee \frac{\sigma_v^2}{L_v} \right)(1 - \tau^{-1})(1+\chi^2)} \right)^{1/2}$$

$$\bigwedge \frac{1}{(1+\chi^2)(1+\rho^2)} \bigwedge \left( \frac{m/n}{(1-\tau^{-1})(1+\rho^2)(1+\chi^2)} \right)^{1/2}.$$

*We have,*

$$\frac{1}{T}\sum_{t=0}^{T-1}\left(\frac{1}{L_u}\mathbf{E}\left\|\nabla_u F\left(u^{(t)},V^{(t)}\right)\right\|^2 + \frac{m}{L_v n^2}\sum_{i=1}^{n}\mathbf{E}\left\|\nabla_v F_i\left(u^{(t)},v_i^{(t)}\right)\right\|^2\right) \leq \varepsilon$$

*after $T$ communication rounds, where, ignoring absolute constants,*

$$T \leq \frac{\Delta F_0(1+\chi^2)}{\varepsilon^2}\left(\frac{\sigma_u^2 + \delta^2\left(1-\frac{m}{n}\right)}{mL_u} + \frac{m\sigma_v^2}{nL_v}\right)$$

$$+ \frac{\Delta F_0\sqrt{(1-\tau^{-1})(1+\chi^2)}}{\varepsilon^{3/2}}\left(\frac{\sigma_u + \delta}{\sqrt{L_u}} + \frac{\sigma_v}{\sqrt{L_v}}\right)$$

$$+ \frac{\Delta F_0}{\varepsilon}(1+\chi^2)(1+\rho^2)\left(1 + \sqrt{\frac{(1-\tau^{-1})n}{m}}\right).$$

*Proof.* The choice of the constant $\eta$ ensures that each of the constant terms in the bound of Theorem 1 is $O(\varepsilon)$. The final rate is now $O\big(\Delta F_0/(\eta\varepsilon)\big)$; plugging in the value of $\eta$ completes the proof. $\square$

We now prove Theorem 1.

*Proof of Theorem 1.* The proof mainly applies the smoothness upper bound to write out a descent condition with suitably small noise terms. We start with some notation.

**Notation.** Let $\mathcal{F}^{(t)}$ denote the $\sigma$-algebra generated by $\left(u^{(t)},V^{(t)}\right)$ and denote $\mathbf{E}_t[\cdot] = \mathbf{E}[\cdot|\mathcal{F}^{(t)}]$. For all devices, including those not selected in each round, we define virtual sequences $\tilde{u}_{i,k}^{(t)},\tilde{v}_{i,k}^{(t)}$ as the SGD updates in Algorithm 4 for all devices regardless of whether they are selected. For the selected devices $k \in S^{(t)}$, we have $\left(u_{i,k}^{(t)},v_{i,k}^{(t)}\right) = \left(\tilde{u}_{i,k}^{(t)},\tilde{v}_{i,k}^{(t)}\right)$. Note now that the random variables $\tilde{u}_{i,k}^{(t)},\tilde{v}_{i,k}^{(t)}$ are independent of the device selection $S^{(t)}$. The updates for the devices $i \in S^{(t)}$ are given by

$$v_i^{(t+1)} = v_i^{(t)} - \gamma_v\sum_{k=0}^{\tau-1}G_{i,v}\left(\tilde{u}_{i,k}^{(t)},\tilde{v}_{i,k}^{(t)},z_{i,k}^{(t)}\right),$$

and the server update is given by

$$u^{(t+1)} = u^{(t)} - \frac{\gamma_u}{m}\sum_{i\in S^{(t)}}\sum_{k=0}^{\tau-1}G_{i,u}\left(\tilde{u}_{i,k}^{(t)},\tilde{v}_{i,k}^{(t)},z_{i,k}^{(t)}\right). \tag{9}$$

**Proof Outline.** We use the smoothness of $F_i$, more precisely Lemma 16, to obtain

$$F\big(u^{(t+1)},V^{(t+1)}\big) - F\big(u^{(t)},V^{(t)}\big)$$

$$\leq \underbrace{\left\langle\nabla_u F(u^{(t)},V^{(t)}),u^{(t+1)}-u^{(t)}\right\rangle}_{\mathcal{T}_{1,u}} + \underbrace{\frac{1}{n}\sum_{i=1}^{n}\left\langle\nabla_v F_i(u^{(t)},v_i^{(t)}),v_i^{(t+1)}-v_i^{(t)}\right\rangle}_{\mathcal{T}_{1,v}} \tag{10}$$

$$+ \underbrace{\frac{L_u(1+\chi^2)}{2}\left\|u^{(t+1)}-u^{(t)}\right\|^2}_{\mathcal{T}_{2,u}} + \underbrace{\frac{1}{n}\sum_{i=1}^{n}\frac{L_v(1+\chi^2)}{2}\left\|v_i^{(t+1)}-v_i^{(t)}\right\|^2}_{\mathcal{T}_{2,v}}.$$

Our goal will be to bound each of these terms to get a descent condition from each step of the form

$$\mathbf{E}_t\left[F\big(u^{(t+1)},V^{(t+1)}\big) - F\big(u^{(t)},V^{(t)}\big)\right]$$

$$\leq -\frac{\gamma_u\tau}{8}\left\|\nabla_u F\big(u^{(t)},V^{(t)}\big)\right\|^2 - \frac{\gamma_v\tau m}{8n^2}\sum_{i=1}^{n}\left\|\nabla_v F_i\big(u^{(t)},v_i^{(t)}\big)\right\|^2 + O(\gamma_u^2 + \gamma_v^2),$$

where the $O(\gamma_u^2 + \gamma_v^2)$ terms are controlled using the bounded variance and gradient diversity assumptions. Telescoping this descent condition gives the final bound.

**Main Proof.** Towards this end, we prove non-asymptotic bounds on each of the terms $\mathcal{T}_{1,v}$, $\mathcal{T}_{1,u}$, $\mathcal{T}_{2,v}$ and $\mathcal{T}_{2,u}$, in Claims 4 to 7 respectively. We then invoke them to get the bound

$$
\begin{aligned}
\mathbf{E}_t \left[ F\big(u^{(t+1)}, V^{(t+1)}\big) - F\big(u^{(t)}, V^{(t)}\big) \right] &\leq -\frac{\gamma_u \tau}{4}\Delta_u^{(t)} - \frac{\gamma_v \tau m}{4n}\Delta_v^{(t)} \\
&+ \frac{L_u(1+\chi^2)\gamma_u^2\tau^2}{2}\left(\sigma_u^2 + \frac{12\delta^2}{m}(1-m/n)\right) + \frac{L_v(1+\chi^2)\gamma_v^2\tau^2\sigma_v^2 m}{2n} \\
&+ \frac{2}{n}\sum_{i=1}^{n}\sum_{k=0}^{\tau-1}\mathbf{E}_t\big\|u_{i,k}^{(t)} - u^{(t)}\big\|^2\left(L_u^2\gamma_u + \frac{m}{n}\chi^2 L_u L_v \gamma_v\right) \\
&+ \frac{2}{n}\sum_{i=1}^{n}\sum_{k=0}^{\tau-1}\mathbf{E}_t\big\|v_{i,k}^{(t)} - v^{(t)}\big\|^2\left(\frac{m}{n}L_v^2\gamma_v + \chi^2 L_u L_v \gamma_u\right).
\end{aligned}
\tag{11}
$$

Note that we simplified some constants appearing on the gradient norm terms using

$$
\gamma_u \leq \big(12 L_u(1+\chi^2)(1+\rho^2)\tau\big)^{-1} \quad \text{and} \quad \gamma_v \leq \big(6 L_v(1+\chi^2)\tau\big)^{-1}.
$$

Our next step is to bound the last two lines of (11) with Lemma 8 and invoke the gradient diversity assumption (Assumption 3') as

$$
\frac{1}{n}\sum_{i=1}^{n}\big\|\nabla_u F_i\big(u^{(t)}, v_i^{(t)}\big)\big\|^2 \leq \delta^2 + (1+\rho^2)\big\|\nabla_u F\big(u^{(t)}, V^{(t)}\big)\big\|^2.
$$

This gives, after plugging in the learning rates and further simplifying the constants,

$$
\begin{aligned}
&\mathbf{E}_t\left[F\big(u^{(t+1)}, V^{(t+1)}\big) - F\big(u^{(t)}, V^{(t)}\big)\right] \\
&\leq -\frac{c\Delta_u^{(t)}}{8L_u} - \frac{cm\Delta_v^{(t)}}{8L_v n} + c^2(1+\chi^2)\left(\frac{\sigma_u^2}{2L_u} + \frac{m\sigma_v^2}{nL_v} + \frac{6\delta^2}{L_u m}\left(1 - \frac{m}{n}\right)\right) \\
&\quad + c^3(1+\chi^2)(1-\tau^{-1})\left(\frac{24\delta^2}{L_u} + \frac{4\sigma_u^2}{L_u} + \frac{4\sigma_v^2}{L_u}\right).
\end{aligned}
$$

Taking full expectation, telescoping the series over $t = 0, \cdots, T-1$ and rearranging the resulting terms give the desired bound in Theorem 1. $\square$

**Claim 4** (Bounding $\mathcal{T}_{1,v}$). *Let $\mathcal{T}_{1,v}$ be defined as in (10). We have,*

$$
\begin{aligned}
\mathbf{E}_t[\mathcal{T}_{1,v}] &\leq -\frac{\gamma_v \tau m}{2n^2}\sum_{i=1}^{n}\big\|\nabla_v F_i\big(u^{(t)}, v_i^{(t)}\big)\big\|^2 \\
&+ \frac{\gamma_v m}{n}\sum_{i=1}^{n}\sum_{k=0}^{\tau-1}\mathbf{E}_t\left[\chi^2 L_u L_v\big\|\tilde{u}_{i,k}^{(t)} - u^{(t)}\big\|^2 + L_v^2\big\|\tilde{v}_{i,k}^{(t)} - v_i^{(t)}\big\|^2\right].
\end{aligned}
$$

*Proof.* Define $\mathcal{T}_{1,v,i}$ to be contribution of the $i$th term to $\mathcal{T}_{1,v}$. For $i \notin S_t$, we have that $\mathcal{T}_{1,v,i} = 0$, since $v_i^{(t+1)} = v_i^{(t)}$. On the other hand, for $i \in S^{(t)}$, we use the unbiasedness of the gradient estimator

$G_{i,v}$ and the independence of $z_{i,k}^{(t)}$ from $u_{i,k}^{(t)}, v_{i,k}^{(t)}$ to get

$$
\begin{aligned}
\mathbf{E}_t\left[\mathcal{T}_{1,v,i}\right] &= -\gamma_v \sum_{k=0}^{\tau-1} \mathbf{E}_t\left\langle \nabla_v F_i\left(u^{(t)}, v_i^{(t)}\right), \nabla_v F_i\left(u_{i,k}^{(t)}, v_{i,k}^{(t)}\right)\right\rangle \\
&= -\gamma_v \sum_{k=0}^{\tau-1} \mathbf{E}_t\left\langle \nabla_v F_i\left(u^{(t)}, v_i^{(t)}\right), \nabla_v F_i\left(\tilde{u}_{i,k}^{(t)}, \tilde{v}_{i,k}^{(t)}\right)\right\rangle \\
&= -\gamma_v \tau \left\|\nabla_v F_i\left(u^{(t)}, v_i^{(t)}\right)\right\|^2 \\
&\quad - \gamma_v \sum_{k=0}^{\tau-1} \mathbf{E}_t\left\langle \nabla_v F_i\left(u^{(t)}, v_i^{(t)}\right), \nabla_v F_i\left(\tilde{u}_{i,k}^{(t)}, \tilde{v}_{i,k}^{(t)}\right) - \nabla_v F_i\left(u^{(t)}, v_i^{(t)}\right)\right\rangle \\
&\le -\frac{\gamma_v \tau}{2}\left\|\nabla_v F_i\left(u^{(t)}, v_i^{(t)}\right)\right\|^2 + \frac{\gamma_v}{2}\sum_{k=0}^{\tau-1}\mathbf{E}_t\left\|\nabla_v F_i\left(\tilde{u}_{i,k}^{(t)}, \tilde{v}_{i,k}^{(t)}\right) - \nabla_v F_i\left(u^{(t)}, v_i^{(t)}\right)\right\|^2. \quad (12)
\end{aligned}
$$

For the second term, we add and subtract $\nabla_v F_i\left(u^{(t)}, \tilde{v}_{i,k}^{(t)}\right)$ and use smoothness to get

$$
\left\|\nabla_v F_i\left(\tilde{u}_{i,k}^{(t)}, \tilde{v}_{i,k}^{(t)}\right) - \nabla_v F_i\left(u^{(t)}, v_i^{(t)}\right)\right\|^2 \le 2\chi^2 L_u L_v\left\|\tilde{u}_{i,k}^{(t)} - u^{(t)}\right\|^2 + 2L_v^2\left\|\tilde{v}_{i,k}^{(t)} - v_i^{(t)}\right\|^2. \tag{13}
$$

Since the right hand side of this bound is independent of $S_t$, we get,

$$
\mathbf{E}_t[\mathcal{T}_{1,v}] = \frac{m}{n}\mathbf{E}_t\left[\frac{1}{m}\sum_{i\in S^{(t)}}\mathcal{T}_{1,v,i}\right] = \frac{m}{n^2}\sum_{i=1}^{n}\mathbf{E}_t[\mathcal{T}_{1,v,i}],
$$

and plugging in (12) and (13) completes the proof. $\qquad\square$

**Claim 5** (Bounding $\mathcal{T}_{1,u}$). *Consider $\mathcal{T}_{1,u}$ defined in* (10). *We have the bound,*

$$
\begin{aligned}
\mathbf{E}_t[\mathcal{T}_{1,u}] \le{}& -\frac{\gamma_u \tau}{2}\left\|\nabla_u F\left(u^{(t)}, V^{(t)}\right)\right\|^2 \\
&+ \frac{\gamma_u}{n}\sum_{i=1}^{n}\sum_{k=0}^{\tau-1}\mathbf{E}_t\left[L_u^2\left\|\tilde{u}_{i,k}^{(t)} - u^{(t)}\right\|^2 + \chi^2 L_u L_v\left\|\tilde{v}_{i,k}^{(t)} - v_i^{(t)}\right\|^2\right].
\end{aligned}
$$

*Proof.* Due to the independence of $S^{(t)}$ from $\tilde{u}_{i,k}^{(t)}, \tilde{v}_{i,k}^{(t)}$, we have,

$$
\begin{aligned}
\mathbf{E}_t\left[u^{(t+1)} - u^{(t)}\right] &= -\gamma_u\mathbf{E}_t\left[\frac{1}{m}\sum_{i\in S^{(t)}}\sum_{k=0}^{\tau-1}\nabla_u F_i\left(u_{i,k}^{(t)}, v_{i,k}^{(t)}\right)\right] \\
&= -\gamma_u\mathbf{E}_t\left[\frac{1}{m}\sum_{i\in S^{(t)}}\sum_{k=0}^{\tau-1}\nabla_u F_i\left(\tilde{u}_{i,k}^{(t)}, \tilde{v}_{i,k}^{(t)}\right)\right] \\
&= -\frac{\gamma_u}{n}\sum_{i=1}^{n}\sum_{k=0}^{\tau-1}\mathbf{E}_t\left[\nabla_u F_i\left(\tilde{u}_{i,k}^{(t)}, \tilde{v}_{i,k}^{(t)}\right)\right],
\end{aligned}
$$

where the last equality took an expectation over $S^{(t)}$, which is independent of $\tilde{u}_{i,k}^{(t)}, \tilde{v}_{i,k}^{(t)}$. Now, using the same sequence of arguments as Claim 4, we have,

$$
\mathbf{E}_t \left\langle \nabla_u F\big(u^{(t)}, V^{(t)}\big), u^{(t+1)} - u^{(t)} \right\rangle
$$

$$
= -\gamma_u \sum_{k=0}^{\tau-1} \mathbf{E}_t \left\langle \nabla_u F\left(u^{(t)}, V^{(t)}\right), \frac{1}{n} \sum_{i=1}^{n} \nabla_u F_i\left(\tilde{u}_{i,k}^{(t)}, \tilde{v}_{i,k}^{(t)}\right) \right\rangle
$$

$$
\leq -\frac{\gamma_u \tau}{2} \left\| \nabla_u F\left(u^{(t)}, V^{(t)}\right) \right\|^2 + \frac{\gamma_u}{2} \sum_{k=0}^{\tau-1} \mathbf{E}_t \left\| \frac{1}{n} \sum_{i=1}^{n} \nabla_u F_i\left(\tilde{u}_{i,k}^{(t)}, \tilde{v}_{i,k}^{(t)}\right) - \nabla_u F\left(u^{(t)}, V^{(t)}\right) \right\|^2
$$

$$
\overset{(*)}{\leq} -\frac{\gamma_u \tau}{2} \left\| \nabla_u F\left(u^{(t)}, V^{(t)}\right) \right\|^2 + \frac{\gamma_u}{2n} \sum_{i=1}^{n} \sum_{k=0}^{\tau-1} \mathbf{E}_t \left\| \nabla_u F_i\left(\tilde{u}_{i,k}^{(t)}, \tilde{v}_{i,k}^{(t)}\right) - \nabla_u F_i\left(u^{(t)}, v_i^{(t)}\right) \right\|^2
$$

$$
\leq -\frac{\gamma_u \tau}{2} \left\| \nabla_u F\left(u^{(t)}, V^{(t)}\right) \right\|^2 + \frac{\gamma_u}{n} \sum_{i=1}^{n} \sum_{k=0}^{\tau-1} \mathbf{E}_t \left[ L_u^2 \left\| \tilde{u}_{i,k}^{(t)} - u^{(t)} \right\|^2 + L_{uv}^2 \left\| \tilde{v}_{i,k}^{(t)} - v_i^{(t)} \right\|^2 \right],
$$

where the inequality $(*)$ follows from Jensen's inequality as

$$
\left\| \frac{1}{n} \sum_{i=1}^{n} \nabla_u F_i\left(\tilde{u}_{i,k}^{(t)}, \tilde{v}_{i,k}^{(t)}\right) - \nabla_u F\left(u^{(t)}, V^{(t)}\right) \right\|^2 \leq \frac{1}{n} \sum_{i=1}^{n} \left\| \nabla_u F_i\left(\tilde{u}_{i,k}^{(t)}, \tilde{v}_{i,k}^{(t)}\right) - \nabla_u F_i\left(u_{i,k}^{(t)}, v^{(t)}\right) \right\|^2.
$$

$\square$

**Claim 6** (Bounding $\mathcal{T}_{2,v}$). *Consider $\mathcal{T}_{2,v}$ as defined in* (10). *We have the bound,*

$$
\mathbf{E}_t[\mathcal{T}_{2,v}] \leq \frac{3 L_v (1+\chi^2) \gamma_v^2 \tau^2 m}{2n^2} \sum_{i=1}^{n} \left\| \nabla_v F_i\left(u^{(t)}, v_i^{(t)}\right) \right\|^2 + \frac{L_v (1+\chi^2) \gamma_v^2 \tau^2 m \sigma_v^2}{2n}
$$

$$
+ \frac{3 L_v (1+\chi^2) \gamma_v^2 \tau m}{2n^2} \sum_{i=1}^{n} \sum_{k=0}^{\tau-1} \mathbf{E}_t \left[ L_v^2 \left\| \tilde{v}_{i,k}^{(t)} - v_i^{(t)} \right\|^2 + \chi^2 L_u L_v \left\| \tilde{u}_{i,k}^{(t)} - u^{(t)} \right\|^2 \right].
$$

*Proof.* We start with

$$
\mathbf{E}_t \left\| \tilde{v}_{k,\tau}^{(t)} - v^{(t)} \right\|^2 = \gamma_v^2 \mathbf{E}_t \left\| \sum_{k=0}^{\tau-1} G_{i,v}\left(\tilde{u}_{i,k}^{(t)}, \tilde{v}_{i,k}^{(t)}, z_{i,k}^{(t)}\right) \right\|^2
$$

$$
\leq \gamma_v^2 \tau \sum_{k=0}^{\tau-1} \mathbf{E}_t \left\| G_{i,v}\left(\tilde{u}_{i,k}^{(t)}, \tilde{v}_{i,k}^{(t)}, z_{i,k}^{(t)}\right) \right\|^2
$$

$$
\leq \gamma_v^2 \tau^2 \sigma_v^2 + \gamma_v^2 \tau \sum_{k=0}^{\tau-1} \mathbf{E}_t \left\| \nabla_v F_i\left(\tilde{u}_{i,k}^{(t)}, \tilde{v}_{i,k}^{(t)}\right) \right\|^2
$$

$$
\leq \gamma_v^2 \tau^2 \sigma_v^2 + 3\gamma_v^2 \tau^2 \left\| \nabla_v F_i\left(u^{(t)}, v_i^{(t)}\right) \right\|^2
$$

$$
+ 3\gamma_v^2 \tau \sum_{k=0}^{\tau-1} \mathbf{E}_t \left[ L_v^2 \left\| \tilde{v}_{i,k}^{(t)} - v_i^{(t)} \right\|^2 + \chi^2 L_u L_v \left\| \tilde{u}_{i,k}^{(t)} - u^{(t)} \right\|^2 \right].
$$

Using (a) $v_i^{(t+1)} = \tilde{v}_{i,\tau}^{(t)}$ for $i \in S^{(t)}$, and, (b) $S^{(t)}$ is independent from $\tilde{u}_{i,k}^{(t)}, \tilde{v}_{i,k}^{(t)}$, we get,

$$
\mathbf{E}_t[\mathcal{T}_{2,v}] = \frac{L_v (1+\chi^2) m}{2n} \mathbf{E}_t \left[ \frac{1}{m} \sum_{i \in S^{(t)}} \left\| \tilde{v}_{i,\tau}^{(t)} - v_i^{(t)} \right\|^2 \right]
$$

$$
\leq \frac{L_v (1+\chi^2) m}{2n^2} \sum_{i=1}^{n} \mathbf{E}_t \left\| \tilde{v}_{i,\tau}^{(t)} - v_i^{(t)} \right\|^2
$$

Plugging in the bound $\mathbf{E}_t \left\| \tilde{v}_{i,\tau}^{(t)} - v^{(t)} \right\|^2$ completes the proof. $\square$

**Claim 7** (Bounding $\mathcal{T}_{2,u}$). *Consider $\mathcal{T}_{2,u}$ as defined in* (10). *We have,*

$$
\begin{aligned}
\mathbf{E}_t[\mathcal{T}_{2,u}] &\leq \frac{L_u(1+\chi^2)\gamma_u^2\tau^2}{2m}\left(\sigma_u^2 + 12\delta^2\left(1 - \frac{m}{n}\right)\right) \\
&\quad + 3L_u(1+\chi^2)\gamma_u^2\tau^2(1+\rho^2)\left\|\nabla_u F_i\left(u^{(t)}, V^{(t)}\right)\right\|^2 \\
&\quad + \frac{3L_u(1+\chi^2)\gamma_u^2\tau}{2n}\sum_{i=1}^{n}\sum_{k=0}^{\tau-1}\mathbf{E}_t\left[L_u^2\left\|\tilde{u}_{i,k}^{(t)} - u^{(t)}\right\|^2 + \chi^2 L_u L_v\left\|\tilde{v}_{i,k}^{(t)} - v_i^{(t)}\right\|^2\right].
\end{aligned}
$$

*Proof.* We proceed with the first two inequalities as in the proof of Claim 6 to get

$$
\mathbf{E}_t\left\|u^{(t+1)} - u^{(t)}\right\|^2 \leq \frac{\gamma_u^2\tau^2\sigma_u^2}{m} + \gamma_u^2\tau\sum_{k=0}^{\tau-1}\underbrace{\mathbf{E}_t\left\|\frac{1}{m}\sum_{i\in S^{(t)}}\nabla_u F_i\left(\tilde{u}_{i,k}^{(t)}, \tilde{v}_{i,k}^{(t)}\right)\right\|^2}_{=:\mathcal{T}_{3,j}}.
$$

For $\mathcal{T}_{3,j}$, (a) we add and subtract $\nabla_u F(u^{(t)}, V^{(t)})$ and $\nabla_u F_i(u^{(t)}, \tilde{v}_{i,k}^{(t)})$, (b) invoke the squared triangle inequality, and, (c) use smoothness to get

$$
\begin{aligned}
\mathcal{T}_{3,j} &= 6\,\mathbf{E}_t\left\|\frac{1}{m}\sum_{i\in S^{(t)}}\nabla_u F_i\left(u^{(t)}, v_i^{(t)}\right) - \nabla_u F\left(u^{(t)}, V^{(t)}\right)\right\|^2 + 6\left\|\nabla_u F\left(u^{(t)}, V^{(t)}\right)\right\|^2 \\
&\quad + 3\mathbf{E}_t\left[\frac{1}{m}\sum_{i\in S^{(t)}}\left(L_u^2\left\|\tilde{u}_{i,k}^{(t)} - u^{(t)}\right\|^2 + \chi^2 L_u L_v\left\|\tilde{v}_{i,k}^{(t)} - v_i^{(t)}\right\|^2\right)\right]
\end{aligned}
$$

For the first term, we use the fact that $S^{(t)}$ is obtained by sampling without replacement to apply Lemma 17 together with the gradient diversity assumption to get

$$
\begin{aligned}
\mathbf{E}_t&\left\|\frac{1}{m}\sum_{i\in S^{(t)}}\nabla_u F_i\left(u^{(t)}, v_i^{(t)}\right) - \nabla_u F\left(u^{(t)}, V^{(t)}\right)\right\|^2 \\
&\leq \frac{1}{m}\left(\frac{n-m}{n-1}\right)\frac{1}{n}\sum_{i=1}^{n}\left\|\nabla_u F_i\left(u^{(t)}, v_i^{(t)}\right) - \nabla_u F\left(u^{(t)}, V^{(t)}\right)\right\|^2 \\
&\leq \frac{1}{m}\left(\frac{n-m}{n-1}\right)\left(\delta^2 + \rho^2\left\|\nabla_u F\left(u^{(t)}, V^{(t)}\right)\right\|^2\right).
\end{aligned}
$$

Therefore,

$$
\begin{aligned}
\mathcal{T}_{3,j} &= \frac{12\delta^2}{m}\left(1 - \frac{m}{n}\right) + 6(1+\rho^2)\left\|\nabla_u F\left(u^{(t)}, V^{(t)}\right)\right\|^2 \\
&\quad + \frac{3}{n}\sum_{i=1}^{n}\mathbf{E}_t\left[L_u^2\left\|\tilde{u}_{i,k}^{(t)} - u^{(t)}\right\|^2 + \chi^2 L_u L_v\left\|\tilde{v}_{i,k}^{(t)} - v_i^{(t)}\right\|^2\right],
\end{aligned}
$$

where we also used the independence between $S^{(t)}$ and $(\tilde{u}_{i,k}^{(t)}, \tilde{v}_{i,k}^{(t)})$. Plugging this into the expression for $\mathbf{E}_t\|u^{(t+1)} - u^{(t)}\|^2$ completes the proof. $\qquad\square$

**Lemma 8.** *Let $F_i$ satisfy Assumptions 1'-3', and consider the iterates*

$$
u_{k+1} = u_k - \gamma_u G_{i,u}(u_k, v_k, z_k), \quad \text{and,} \quad v_{k+1} = v_k - \gamma_v G_{i,v}(u_k, v_k, z_k),
$$

*for $k = 0, \cdots, \tau-1$, where $z_k \sim \mathcal{D}_i$. Suppose the learning rates satisfy $\gamma_u = c_u/(\tau L_u)$ and $\gamma_v = c_v/(\tau L_v)$ with $c_u, c_v \leq 1/\sqrt{6}\max\{1, \chi^{-2}\}$. Further, define,*

$$
A = \gamma_u L_u^2 + f\chi^2\gamma_v L_u L_v, \quad \text{and,} \quad B = f\gamma_v L_v^2 + \chi^2\gamma_u L_u L_v,
$$

*where $f \in (0, 1]$ is given. Then, we have the bound,*

$$\sum_{k=0}^{\tau-1} \mathbf{E}\big[A\|u_k - u_0\|^2 + B\|v_k - u_0\|^2\big] \leq 4\tau^2(\tau-1)\left(\gamma_u^2 \sigma_u^2 A + \gamma_v^2 \sigma_v^2 B\right)$$
$$+ 12\tau^2(\tau-1)\left(\gamma_u^2 A\|\nabla_u F_i(u_0, v_0)\|^2 + \gamma_v^2 B\|\nabla_v F_i(u_0, v_0)\|^2\right).$$

*Proof.* If $\tau = 1$, there is nothing to prove, so we assume $\tau > 1$. Let $\Delta_k := A\|u_k - u_0\|^2 + B\|v_k - v_0\|^2$ and denote by $\mathcal{F}_k$ the sigma-algebra generated by $(w_k, v_k)$. Further, let $\mathbf{E}_k[\cdot] = \mathbf{E}[\cdot|\mathcal{F}_k]$. We use the inequality $2\alpha\beta \leq \alpha^2/\delta^2 + \delta^2\beta^2$ for reals $\alpha, \beta, \delta$ to get,

$$\mathbf{E}_k\|u_{k+1} - u_0\|^2 \leq \left(1 + \frac{1}{\tau-1}\right)\|u_k - u_0\|^2 + \tau\gamma_u^2\mathbf{E}_k\|G_{i,u}(u_k, v_k, z_k)\|^2$$
$$\leq \left(1 + \frac{1}{\tau-1}\right)\|u_k - u_0\|^2 + \tau\gamma_u^2\sigma_u^2 + \tau\gamma_u^2\|\nabla_u F_i(u_k, v_k)\|^2$$
$$\leq \left(1 + \frac{1}{\tau-1}\right)\|u_k - u_0\|^2 + \tau\gamma_u^2\sigma_u^2 + 3\tau\gamma_u^2\|\nabla_u F_i(u_0, v_0)\|^2$$
$$+ 3\tau\gamma_u^2 L_u^2\|u_k - u_0\|^2 + 3\tau\gamma_u^2 L_{uv}\|v_k - v_0\|^2,$$

where the last inequality followed from the squared triangle inequality (from adding and subtracting $\nabla_u F_i(u_0, v_k)$ and $\nabla_u F_i(u_0, v_0)$) followed by smoothness. Together with the analogous inequality for the $v$-update, we get,

$$\mathbf{E}_k[\Delta_{k+1}] \leq \left(1 + \frac{1}{\tau-1}\right)\Delta_k + A'\|u_k - u_0\|^2 + B'\|v_k - v_0\|^2 + C,$$

where we have

$$A' = 3\tau(\gamma_u^2 L_u^2 A + \gamma_v^2 \chi^2 L_u L_v B), \quad \text{and,} \quad B' = 3\tau(\gamma_v^2 L_v^2 B + \gamma_u^2 \chi^2 L_u L_v A) \quad \text{and,}$$
$$C' = \tau\gamma_u^2\sigma_u^2 A + \tau\gamma_v^2\sigma_v^2 B + 3\tau\gamma_u^2 A\|\nabla_u F_i(u_0, v_0)\|^2 + 3\tau\gamma_v^2 B\|\nabla_v F_i(u_0, v_0)\|^2.$$

Next, we apply Lemma 20 to get that $A' \leq A/\tau$ and $B' \leq B/\tau$ under the assumed conditions on the learning rates; this allows us to write the right hand side completely in terms of $\Delta_k$ and unroll the recurrence. The intuition behind Lemma 20 is as follows. Ignoring the dependence on $\tau, L_u, L_v, \chi$ for a moment, if $\gamma_u$ and $\gamma_v$ are both $O(\eta)$, then $A', B'$ are both $O(\eta^3)$, while $A$ and $B$ are $O(\eta)$. Thus, making $\eta$ small enough should suffice to get $A' \leq O(A)$ and $B' \leq O(B)$.

Concretely, Lemma 20 gives

$$\mathbf{E}_k[\Delta_{k+1}] \leq \left(1 + \frac{2}{\tau-1}\right)\mathbf{E}[\Delta_k] + C,$$

and unrolling this recurrence gives for $k \leq \tau - 1$

$$\mathbf{E}[\Delta_k] \leq \sum_{j=0}^{k-1}\left(1 + \frac{2}{\tau-1}\right)^j C \leq \frac{\tau-1}{2}\left(1 + \frac{2}{\tau-1}\right)^k C$$
$$\leq \frac{\tau-1}{2}\left(1 + \frac{2}{\tau-1}\right)^{\tau-1} C \leq \frac{e^2}{2}(\tau-1)C,$$

where we used $(1 + 1/\alpha)^\alpha \leq e$ for all $\alpha > 0$. Summing over $k$ and using the numerical bound $e^2 < 8$ completes the proof. □

**Remark 9.** *We only invoked the partial gradient diversity assumption (Assumption 3) at iterates $(u^{(t)}, V^{(t)})$; therefore, it suffices if the assumption only holds at iterates $(u^{(t)}, V^{(t)})$ generated by FedSim, rather than at all $(u, V)$.*

---

**Algorithm 5** FedAlt: Alternating updates of shared and personalized parameters

---

**Input:** Initial iterates $u^{(0)}, V^{(0)}$, Number of communication rounds $T$, Number of devices per round $m$, Number of local updates $\tau_u, \tau_v$, Local step sizes $\gamma_u, \gamma_v$,

1: **for** $t = 0, 1, \cdots, T-1$ **do**
2:      Sample $m$ devices from $[n]$ without replacement in $S^{(t)}$
3:      **for** each selected device $i \in S^{(t)}$ in parallel **do**
4:          Initialize $v_{i,0}^{(t)} = v_i^{(t)}$
5:          **for** $k = 0, \cdots, \tau_v - 1$ **do**                    $\triangleright$ Update personalized parameters
6:             Sample data $z_{i,k}^{(t)} \sim \mathcal{D}_i$
7:             $v_{i,k+1}^{(t)} = v_{i,k}^{(t)} - \gamma_v G_{i,v}(u^{(t)}, v_{i,k}^{(t)}, z_{i,k}^{(t)})$
8:          Update $v_i^{(t+1)} = v_{k,\tau_v}^{(t)}$
9:          Initialize $u_{i,0}^{(t)} = u^{(t)}$
10:          **for** $k = 0, \cdots, \tau_u - 1$ **do**                   $\triangleright$ Update shared parameters
11:             $u_{i,k+1}^{(t)} = u_{i,k}^{(t)} - \gamma_u G_{i,u}(u_{i,k}^{(t)}, v_i^{(t+1)}, z_{i,k}^{(t)})$
12:          Update $u_i^{(t+1)} = u_{i,\tau_u}^{(t)}$
13:      Update $u^{(t+1)} = \frac{\sum_{i \in S^{(t)}} \alpha_i u_i^{(t+1)}}{\sum_{i \in S^{(t)}} \alpha_i}$ at the server with secure aggregation
14: **return** $u^{(T)}, v_1^{(T)}, \cdots, v_n^{(T)}$

---

## A.3   CONVERGENCE ANALYSIS OF FEDALT

We give the full form of FedAlt in Algorithms 5 for the general case of unequal $\alpha_i$'s but focus on $\alpha_i = 1/n$ for the analysis. For convenience, we reiterate Theorem 2 below. Recall the definitions

$$\Delta_u^{(t)} = \left\| \nabla_u F\left(u^{(t)}, V^{(t+1)}\right) \right\|^2, \quad \text{and,} \quad \Delta_v^{(t)} = \frac{1}{n}\sum_{i=1}^n \left\| \nabla_v F_i\left(u^{(t)}, v_i^{(t)}\right) \right\|^2.$$

**Theorem 2 (Convergence of FedAlt).** *Suppose Assumptions 1, 2 and 3 hold and the learning rates in FedAlt are chosen as $\gamma_u = \eta/(L_u \tau_u)$ and $\gamma_v = \eta/(L_v \tau_v)$, with*

$$\eta \le \min\left\{ \frac{1}{24(1+\rho^2)}, \frac{m}{128\chi^2(n-m)}, \sqrt{\frac{m}{\chi^2 n}} \right\}.$$

*Then, ignoring absolute constants, we have*

$$\frac{1}{T}\sum_{t=0}^{T-1}\left( \frac{1}{L_u}\mathbf{E}[\Delta_u^{(t)}] + \frac{m}{nL_v}\mathbf{E}[\Delta_v^{(t)}] \right) \le \frac{\Delta F_0}{\eta T} + \eta\left( \frac{\sigma_u^2 + \delta^2(1 - \frac{m}{n})}{mL_u} + \frac{\sigma_v^2}{L_v}\frac{m + \chi^2(n-m)}{n} \right)$$
$$+ \eta^2\left( \frac{\sigma_u^2 + \delta^2}{L_u}(1 - \tau_u^{-1}) + \frac{\sigma_v^2 m}{L_v n}(1 - \tau_v^{-1}) + \frac{\chi^2 \sigma_v^2}{L_v} \right).$$

Before proving the theorem, we have the corollary with optimized learning rates.

**Corollary 10.** *Consider the setting of Theorem 2 and fix some $\varepsilon > 0$. Suppose we set $\gamma_u = \eta/(\tau L_u)$ and $\gamma_v = \eta/(\tau L_v)$ such that, ignoring absolute constants,*

$$\eta = \left( \frac{\sigma_v^2}{\varepsilon L_v}\left( \frac{m}{n} + \chi^2(1 - m/n) \right) \right)^{-1} \bigwedge \left( \frac{\sigma_u^2 + \delta^2(1 - m/n)}{mL_u \varepsilon} \right)^{-1} \bigwedge \left( \frac{\sigma_u^2 + \delta^2}{L_u \varepsilon}(1 - \tau_u^{-1}) \right)^{-1/2}$$
$$\bigwedge \left( \frac{\sigma_v^2 m}{L_v n \varepsilon}(1 - \tau_v^{-1}) \right)^{-1/2} \bigwedge \frac{1}{1+\rho^2} \bigwedge \frac{m}{\chi^2(n-m)} \bigwedge \sqrt{\frac{m}{\chi^2 n}}.$$

*Then, we have,*

$$\frac{1}{T}\sum_{t=0}^{T-1}\left( \frac{1}{L_u}\mathbf{E}\left\| \nabla_u F\left(u^{(t)}, \tilde{V}^{(t)}\right) \right\|^2 + \frac{m}{L_v n^2}\sum_{i=1}^n \mathbf{E}\left\| \nabla_v F_i\left(u^{(t)}, v_i^{(t)}\right) \right\|^2 \right) \le \varepsilon$$

*after $T$ communication rounds, where, ignoring absolute constants,*

$$T \leq \frac{\Delta F_0}{\varepsilon^2} \left( \frac{\sigma_u^2 + \delta^2 \left(1 - \frac{m}{n}\right)}{m L_u} + \frac{\sigma_v^2}{L_v} \left( \frac{m}{n} + \chi^2 \left(1 - \frac{m}{n}\right) \right) \right)$$

$$+ \frac{\Delta F_0}{\varepsilon^{3/2}} \left( \frac{\sigma_u + \delta}{\sqrt{L_u}} \sqrt{1 - \tau_u^{-1}} + \frac{\sigma_v}{\sqrt{L_v}} \sqrt{1 - \tau_v^{-1}} \right)$$

$$+ \frac{\Delta F_0}{\varepsilon} \left( 1 + \rho^2 + \chi^2 \left( \frac{n}{m} - 1 \right) + \sqrt{\frac{\chi^2 n}{m}} \right).$$

*Proof.* We get the bound by balancing terms from the bound of Theorem 2. The choice of $\eta$ ensures that all the $O(\eta)$ and $O(\eta^2)$ terms are at most $O(\varepsilon)$. Finally, the smallest number of communication rounds to make the left hand side of the bound of Theorem 2 smaller than $\varepsilon$ is $\Delta F_0 / (\eta \varepsilon)$. $\qquad\square$

We are now ready to prove Theorem 2.

*Proof of Theorem 2.* The proof mainly applies the smoothness upper bound to write out a descent condition with suitably small noise terms. We start with some notation.

We introduce the notation $\widetilde{\Delta}_u^{(t)}$ as the analogue of $\Delta_u^{(t)}$ with the virtual variable $\widetilde{V}^{(t+1)}$:

$$\widetilde{\Delta}_u^{(t)} = \left\| \nabla_u F \left( u^{(t)}, V^{(t)} \right) \right\|^2.$$

**Notation.** Let $\mathcal{F}^{(t)}$ denote the $\sigma$-algebra generated by $\left( u^{(t)}, V^{(t)} \right)$ and denote $\mathbf{E}_t[\cdot] = \mathbf{E}[\cdot | \mathcal{F}^{(t)}]$. For all devices, including those not selected in each round, we define virtual sequences $\tilde{u}_{i,k}^{(t)}, \tilde{v}_{i,k}^{(t)}$ as the SGD updates in Algorithm 5 for all devices regardless of whether they are selected. For the selected devices $i \in S^{(t)}$, we have $v_{i,k}^{(t)} = \tilde{v}_{i,k}^{(t)}$ and $u_{i,k}^{(t)} = \tilde{u}_{i,k}^{(t)}$. Note now that the random variables $\tilde{u}_{i,k}^{(t)}, \tilde{v}_{i,k}^{(t)}$ are independent of the device selection $S^{(t)}$. Finally, we have that the updates for the selected devices $i \in S^{(t)}$ are given by

$$v_i^{(t+1)} = v_i^{(t)} - \gamma_v \sum_{k=0}^{\tau_v - 1} G_{i,v} \left( u^{(t)}, \tilde{v}_{i,k}^{(t)}, z_{i,k}^{(t)} \right),$$

and the server update is given by

$$u^{(t+1)} = u^{(t)} - \frac{\gamma_u}{m} \sum_{i \in S^{(t)}} \sum_{k=0}^{\tau_u - 1} G_{i,u} \left( \tilde{u}_{i,k}^{(t)}, \tilde{v}_{i,\tau_v}^{(t)}, z_{i,k}^{(t)} \right).$$

**Proof Outline and the Challenge of Dependent Random Variables.** We start with

$$\begin{aligned} F\left( u^{(t+1)}, V^{(t+1)} \right) - F\left( u^{(t)}, V^{(t)} \right) &= F\left( u^{(t)}, V^{(t+1)} \right) - F\left( u^{(t)}, V^{(t)} \right) \\ &\quad + F\left( u^{(t+1)}, V^{(t+1)} \right) - F\left( u^{(t)}, V^{(t+1)} \right). \end{aligned} \tag{14}$$

The first line corresponds to the effect of the $v$-step and the second line to the $u$-step. The former is easy to handle with standard techniques that rely on the smoothness of $F\left( u^{(t)}, \cdot \right)$. The latter is more challenging. In particular, the smoothness bound for the $u$-step gives us

$$F\left( u^{(t+1)}, V^{(t+1)} \right) - F\left( u^{(t)}, V^{(t+1)} \right)$$

$$\leq \left\langle \nabla_u F \left( u^{(t)}, V^{(t+1)} \right), u^{(t+1)} - u^{(t)} \right\rangle + \frac{L_u}{2} \left\| u^{(t+1)} - u^{(t)} \right\|^2.$$

The standard proofs of convergence of stochastic gradient methods rely on the fact that we can take an expectation w.r.t. the sampling $S^{(t)}$ of devices for the first order term. However, both $V^{(t+1)}$ and

$u^{(t+1)}$ depend on the sampling $S^{(t)}$ of devices. Therefore, we cannot directly take an expectation with respect to the sampling of devices in $S^{(t)}$.

**Virtual Full Participation to Circumvent Dependent Random Variables.** The crux of the proof lies in replacing $V^{(t+1)}$ in the analysis of the $u$-step with the virtual iterate $\widetilde{V}^{(t+1)}$ so as to move all the dependence of the $u$-step on $S^{(t)}$ to the $u^{(t+1)}$ term. This allows us to take an expectation; it remains to carefully bound the resulting error terms.

Finally, we will arrive at a bound of the form

$$\frac{1}{T}\sum_{t=0}^{T-1}\left(\frac{\gamma_u\tau_u}{8}\mathbf{E}[\widetilde{\Delta}_u^{(t)}] + \frac{\gamma_v\tau_v m}{16n}\mathbf{E}[\Delta_v^{(t)}]\right) \le \frac{\Delta F_0}{T} + O(\gamma_u^2 + \gamma_v^2).$$

Next, we translate this bound from gradient $\mathbf{E}[\widetilde{\Delta}_u^{(t)}]$ of the virtual $\widetilde{V}^{(t+1)}$ to $\mathbf{E}[\Delta_u^{(t)}]$, which is the gradient computed at the actual iterate $V^{(t)}$. A careful analysis shows that we only incur a lower order term of $O(\gamma_u\gamma_v^2)$ in this translation. Choosing $\gamma_u$ and $\gamma_v$ small enough will give us the final result.

**Analysis of the $u$-Step with Virtual Full Participation.** We introduce the virtual iterates $\widetilde{V}^{(t+1)}$ into the analysis of the $u$-step as follows:

$$F\left(u^{(t+1)}, V^{(t+1)}\right) - F\left(u^{(t)}, V^{(t+1)}\right)$$

$$\le \left\langle \nabla_u F\left(u^{(t)}, V^{(t+1)}\right), u^{(t+1)} - u^{(t)}\right\rangle + \frac{L_u}{2}\left\|u^{(t+1)} - u^{(t)}\right\|^2$$

$$= \left\langle \nabla_u F\left(u^{(t)}, \widetilde{V}^{(t+1)}\right), u^{(t+1)} - u^{(t)}\right\rangle + \frac{L_u}{2}\left\|u^{(t+1)} - u^{(t)}\right\|^2$$

$$\quad + \left\langle \nabla_u F\left(u^{(t)}, V^{(t+1)}\right) - \nabla_u F\left(u^{(t)}, \widetilde{V}^{(t+1)}\right), u^{(t+1)} - u^{(t)}\right\rangle$$

$$\le \left\langle \nabla_u F\left(u^{(t)}, \widetilde{V}^{(t+1)}\right), u^{(t+1)} - u^{(t)}\right\rangle + L_u\left\|u^{(t+1)} - u^{(t)}\right\|^2$$

$$\quad + \frac{1}{2L_u}\left\|\nabla_u F\left(u^{(t)}, V^{(t+1)}\right) - \nabla_u F\left(u^{(t)}, \widetilde{V}^{(t+1)}\right)\right\|^2$$

$$\le \underbrace{\left\langle \nabla_u F\left(u^{(t)}, \widetilde{V}^{(t+1)}\right), u^{(t+1)} - u^{(t)}\right\rangle}_{\mathcal{T}_{1,u}} + \underbrace{L_u\left\|u^{(t+1)} - u^{(t)}\right\|^2}_{\mathcal{T}_{2,u}} + \underbrace{\frac{\chi^2 L_v}{2n}\sum_{i=1}^{n}\left\|\tilde{v}_i^{(t+1)} - v_i^{(t+1)}\right\|^2}_{\mathcal{T}_{3,u}}.$$

The last two inequalities follow from Young's inequality and Lipschitzness of $V \mapsto \nabla_u F(u, V)$ respectively.

We have now successfully eliminated the dependence of the first-order term $\mathcal{T}_{1,u}$ on $V^{(t+1)}$. The virtual iterates $\widetilde{V}^{(t+1)}$ are now independent of $S^{(t)}$. This allows us to take an expectation w.r.t. the sampling $S^{(t)}$ of the devices.

We bound each of these terms in Claims 11 to 13 below to get

$$\mathbf{E}_t\left[F\left(u^{(t+1)}, V^{(t+1)}\right) - F\left(u^{(t)}, V^{(t+1)}\right)\right]$$

$$\le -\frac{\gamma_u\tau_u}{4}\mathbf{E}_t[\widetilde{\Delta}_u^{(t)}] + \underbrace{\frac{2\gamma_u L_u^2}{n}\sum_{i=1}^{n}\sum_{k=0}^{\tau_u-1}\mathbf{E}_t\left\|\tilde{u}_{i,k}^{(t)} - u^{(t)}\right\|^2}_{=:\mathcal{T}_{2,u}'} + 4\gamma_v^2\tau_v^2 L_v\sigma_v^2\chi^2(1-m/n)$$

$$\quad + \frac{L_u\gamma_u^2\tau_u^2}{m}\left(\sigma_u^2 + 3\delta^2\left(1 - \frac{m}{n}\right)\right) + 8\gamma_v^2\tau_v^2 L_v\chi^2(1-m/n)\Delta_v^{(t)}.$$

Note that we used the fact that $24L_u\gamma_u\tau_u(1+\rho^2) \le 1$ to simply the coefficients of some of the terms above. The second term has also been referred to as client drift in the literature; we bound it with

Lemma 18 and invoke the assumption on gradient diversity (Assumption $3'$) to get

$$
\begin{aligned}
\mathcal{T}_{2,u}' &\le \frac{16\gamma_u^3 L_u^2 \tau_u (\tau_u - 1)}{n} \sum_{i=1}^{n} \mathbf{E}_t \left\| \nabla_u F_i \left( u^{(t)}, \tilde{v}_i^{(t+1)} \right) \right\|^2 + 8\gamma_u^3 L_u^2 \tau_u^2 (\tau_u - 1) \sigma_u^2 \\
&\le \frac{16\gamma_u^3 L_u^2 \tau_u (\tau_u - 1)}{n} \left( \delta^2 + \rho^2 \mathbf{E}_t \left\| \nabla_u F \left( u^{(t)}, \widetilde{V}^{(t+1)} \right) \right\|^2 \right) + 8\gamma_u^3 L_u^2 \tau_u^2 (\tau_u - 1) \sigma_u^2.
\end{aligned}
$$

Plugging this back in, we get,

$$
\begin{aligned}
\mathbf{E}_t & \left[ F \left( u^{(t+1)}, V^{(t+1)} \right) - F \left( u^{(t)}, V^{(t+1)} \right) \right] \\
&\le -\frac{\gamma_u \tau_u}{8} \mathbf{E}_t [\widetilde{\Delta}_u^{(t)}] + \frac{L_u \gamma_u^2 \tau_u^2}{m} \left( \sigma_u^2 + 2\delta^2 (1 - m/n) \right) + 4\gamma_v^2 \tau_v^2 L_v \sigma_v^2 \chi^2 (1 - m/n) \\
&\quad + 8\gamma_v^2 \tau_v^2 L_v \chi^2 (1 - m/n) \Delta_v^{(t)} + 8\gamma_u^2 L_u^3 \tau_u^2 (\tau_u - 1)(\sigma_u^2 + 2\delta_u^2).
\end{aligned}
$$

Note that we used $128\gamma_u^2 L_u^2 \tau_u (\tau_u - 1) \rho^2 \le 1$, which is implied by $24 L_u \gamma_u \tau_u (1 + \rho^2) \le 1$.

**Bound with the Virual Iterates.** We plug this analysis of the $u$-step and Claim 14 for the $v$-step into (14) next. We also simplify some coefficients using $128\gamma_v \tau_v L_v \chi^2 (n/m - 1) \le 1$. This gives us

$$
\begin{aligned}
\mathbf{E}_t & \left[ F \left( u^{(t+1)}, V^{(t+1)} \right) - F \left( u^{(t)}, V^{(t)} \right) \right] \\
&\le -\frac{\gamma_u \tau_u}{8} \mathbf{E}_t [\widetilde{\Delta}_u^{(t)}] - \frac{\gamma_v \tau_v m}{16n} \mathbf{E}_t [\Delta_v^{(t)}] + 4\gamma_v^2 L_v \tau_v^2 \sigma_v^2 \left( \frac{m}{n} + \chi^2 (1 - m/n) \right) \\
&\quad + \frac{\gamma_u^2 L_u \tau_u^2}{m} \left( \sigma_u^2 + 2\delta^2 (1 - m/n) \right) + 8\gamma_u^3 L_u^2 \tau_u^2 (\tau_u - 1)(\sigma_u^2 + 2\delta^2) + \frac{4\gamma_v^3 L_v^2 \tau_v^2 (\tau_v - 1) \sigma_v^2 m}{n}.
\end{aligned}
$$

Taking an unconditional expectation, summing it over $t = 0$ to $T - 1$ and rearranging this gives

$$
\begin{aligned}
\frac{1}{T} \sum_{t=0}^{T-1} & \left( \frac{\gamma_u \tau_u}{8} \mathbf{E}[\widetilde{\Delta}_u^{(t)}] + \frac{\gamma_v \tau_v m}{16n} \mathbf{E}[\Delta_v^{(t)}] \right) \quad\quad\quad\quad\quad\quad\quad\quad\quad\quad (15) \\
&\le \frac{\Delta F_0}{T} + 4\gamma_v^2 L_v \tau_v^2 \sigma_v^2 \left( \frac{m}{n} + \chi^2 (1 - m/n) \right) + \frac{\gamma_u^2 L_u \tau_u^2}{m} \left( \sigma_u^2 + 2\delta^2 (1 - m/n) \right) \\
&\quad + 8\gamma_u^3 L_u^2 \tau_u^2 (\tau_u - 1)(\sigma_u^2 + 2\delta^2) + \frac{4\gamma_v^3 L_v^2 \tau_v^2 (\tau_v - 1) \sigma_v^2 m}{n}.
\end{aligned}
$$

This is a bound in terms of the virtual iterates $\widetilde{V}^{(t+1)}$. However, we wish to show a bound in terms of the actual iterate $V^{(t)}$.

**Obtaining the Final Bound.** It remains now to relate $\widetilde{\Delta}_u^{(t)}$ with $\Delta_u^{(t)}$. Using the Cauchy-Schwartz inequality and smoothness, we have,

$$
\begin{aligned}
\mathbf{E}_t \left\| \nabla_u F \left( u^{(t)}, V^{(t)} \right) - \nabla_u F \left( u^{(t)}, \widetilde{V}^{(t+1)} \right) \right\|^2 & \\
&\le \frac{1}{n} \sum_{i=1}^{n} \mathbf{E}_t \left\| \nabla_u F_i \left( u^{(t)}, v_i^{(t)} \right) - \nabla_u F_i \left( u^{(t)}, \tilde{v}_i^{(t+1)} \right) \right\|^2 \\
&\le \frac{\chi^2 L_u L_v}{n} \sum_{i=1}^{n} \mathbf{E}_t \left\| \tilde{v}_i^{(t+1)} - v_i^{(t)} \right\|^2 \\
&\le \frac{\chi^2 L_u L_v}{n} \sum_{i=1}^{n} \left( 16\gamma_v^2 \tau_v^2 \left\| \nabla_v F_i \left( u^{(t)}, v_i^{(t)} \right) \right\|^2 + 8\gamma_v^2 \tau_v^2 \sigma_v^2 \right) \\
&= 8\gamma_v^2 \tau_v^2 \sigma_v^2 \chi^2 L_u L_v + 16\gamma_v^2 \tau_v^2 \chi^2 L_u L_v \Delta_v^{(t)},
\end{aligned}
$$

where the last inequality followed from Lemma 19. Using

$$
\left\| \nabla_u F \left( u^{(t)}, V^{(t)} \right) \right\|^2 \le 2 \left\| \nabla_u F \left( u^{(t)}, V^{(t)} \right) - \nabla_u F \left( u^{(t)}, \widetilde{V}^{(t+1)} \right) \right\|^2 + 2 \left\| \nabla_u F \left( u^{(t)}, \widetilde{V}^{(t+1)} \right) \right\|^2,
$$

we get,
$$\mathbf{E}[\Delta_u^{(t)}] \le 2\,\mathbf{E}[\widetilde{\Delta}_u^{(t)}] + 16\gamma_v^2\tau_v^2\sigma_v^2\chi^2 L_u L_v + 32\gamma_v^2\tau_v^2\chi^2 L_u L_v\,\mathbf{E}[\Delta_v^{(t)}]\,.$$

Therefore, we get,

$$\frac{\gamma_u\tau_u}{16}\mathbf{E}[\Delta_u^{(t)}] + \frac{\gamma_v\tau_v m}{32n}\mathbf{E}[\Delta_v^{(t)}]$$

$$\le \frac{\gamma_u\tau_u}{8}\mathbf{E}[\widetilde{\Delta}_u^{(t)}] + \frac{\gamma_v\tau_v m}{16n}\left(\frac{1}{2} + \frac{32\eta^2\chi^2 m}{n}\right)\mathbf{E}[\Delta_v^{(t)}] + \gamma_u\tau_u\gamma_v^2\tau_v^2\sigma_v^2\chi^2 L_u L_v$$

$$\le \frac{\gamma_u\tau_u}{8}\mathbf{E}[\widetilde{\Delta}_u^{(t)}] + \frac{\gamma_v\tau_v m}{16n}\mathbf{E}[\Delta_v^{(t)}] + \gamma_u\tau_u\gamma_v^2\tau_v^2\sigma_v^2\chi^2 L_u L_v\,,$$

where we used $\frac{32\eta^2\chi^2 m}{n} \le 1/2$, which is one of the conditions we assume on $\eta$.

Summing this up and plugging in (15) gives

$$\frac{1}{T}\sum_{t=0}^{T-1}\left(\frac{\gamma_u\tau_u}{16}\mathbf{E}[\Delta_u^{(t)}] + \frac{\gamma_v\tau_v m}{32n}\mathbf{E}[\Delta_v^{(t)}]\right)$$

$$\le \frac{1}{T}\sum_{t=0}^{T-1}\left(\frac{\gamma_u\tau_u}{8}\mathbf{E}[\widetilde{\Delta}_u^{(t)}] + \frac{\gamma_v\tau_v m}{16n}\mathbf{E}[\Delta_v^{(t)}]\right) + \gamma_u\tau_u\gamma_v^2\tau_v^2\sigma_v^2\chi^2 L_u L_v$$

$$\le \frac{\Delta F_0}{T} + 4\gamma_v^2 L_v\tau_v^2\sigma_v^2\left(\frac{m}{n} + \chi^2(1 - m/n)\right) + \frac{\gamma_u^2 L_u\tau_u^2}{m}\left(\sigma_u^2 + 2\delta^2(1 - m/n)\right)$$

$$+ 8\gamma_u^3 L_u^2\tau_u^2(\tau_u - 1)(\sigma_u^2 + 2\delta^2) + \frac{4\gamma_v^3 L_v^2\tau_v^2(\tau_v - 1)\sigma_v^2 m}{n} + \gamma_u\tau_u\gamma_v^2\tau_v^2\sigma_v^2\chi^2 L_u L_v\,.$$

Plugging in $\gamma_u = \eta/(L_u\tau_u)$ and $\gamma_v = \eta/(L_v\tau_v)$ completes the proof. $\qquad\square$

The analysis of each of the terms in the $u$-step is given in the following claims.

**Claim 11** (Bounding $\mathcal{T}_{1,u}$). *We have,*

$$\mathbf{E}_t\left[\mathcal{T}_{1,u}\right] \le -\frac{\gamma_u\tau_u}{2}\mathbf{E}_t\left\|\nabla_u F\left(u^{(t)}, \widetilde{V}^{(t+1)}\right)\right\|^2 + \frac{\gamma_u L_u^2}{n}\sum_{i=1}^{n}\sum_{k=0}^{\tau_u-1}\mathbf{E}_t\left\|\tilde{u}_{i,k}^{(t)} - u^{(t)}\right\|^2\,.$$

*Proof.* For $i \in S^{(t)}$, we have that $\tilde{u}_{i,k}^{(t)} = u_{i,k}^{(t)}$. Therefore, we have,

$$\mathbf{E}_t[\mathcal{T}_{1,u}] = -\gamma_u\mathbf{E}_t\left\langle\nabla_u F\left(u^{(t)}, \widetilde{V}^{(t+1)}\right), \frac{1}{m}\sum_{i \in S^{(t)}}\sum_{k=0}^{\tau_u-1}\nabla_u F_i\left(\tilde{u}_{i,k}^{(t)}, \tilde{v}_i^{(t+1)}\right)\right\rangle\,.$$

Using that $\tilde{u}_{i,k}^{(t)}$ is independent of $S^{(t)}$, we get,

$$\mathbf{E}_t[\mathcal{T}_{1,u}] = -\gamma_u\mathbf{E}_t\left\langle\nabla_u F\left(u^{(t)}, \widetilde{V}^{(t+1)}\right), \frac{1}{n}\sum_{i=1}^{n}\sum_{k=0}^{\tau_u-1}\nabla_u F_i\left(\tilde{u}_{i,k}^{(t)}, \tilde{v}_i^{(t+1)}\right)\right\rangle$$

$$= -\gamma_u\tau_u\mathbf{E}_t\left\|\nabla_u F\left(u^{(t)}, \widetilde{V}^{(t+1)}\right)\right\|^2$$

$$- \gamma_u\sum_{k=0}^{\tau_u-1}\mathbf{E}_t\left\langle\nabla_u F\left(u^{(t)}, \widetilde{V}^{(t+1)}\right), \frac{1}{n}\sum_{i=1}^{n}\nabla_u F_i\left(\tilde{u}_{i,k}^{(t)}, \tilde{v}^{(t+1)}\right) - \nabla_u F_i\left(u^{(t)}, \tilde{v}^{(t+1)}\right)\right\rangle$$

Invoking $\langle x, y\rangle \le \|x\|^2/2 + \|y\|^2/2$ for vectors $x, y$ followed by smoothness completes the proof. $\quad\square$

**Claim 12** (Bounding $\mathcal{T}_{2,u}$). *We have,*

$$\mathbf{E}_t\left[\mathcal{T}_{2,u}\right] \le 3L_u\gamma_u^2\tau_u^2\left(1 + \frac{2\rho^2}{m}(1 - m/n)\right)\mathbf{E}_t\left\|\nabla_u F\left(u^{(t)}, \widetilde{V}^{(t+1)}\right)\right\|^2$$

$$+ \frac{3L_u^2\gamma_u^2\tau_u}{n}\sum_{i=1}^{n}\sum_{k=0}^{\tau_u-1}\mathbf{E}_t\left\|\tilde{u}_{i,k}^{(t)} - u^{(t)}\right\|^2 + \frac{6L_u\gamma_u^2\tau_u^2\delta^2}{m}(1 - m/n)\,.$$

*Proof.* We use $\mathbf{E}\|z\|^2 = \|\mathbf{E}[z]\|^2 + \mathbf{E}\|z - \mathbf{E}[z]\|^2$ for a random vector $z$ to get

$$\mathbf{E}_t[\mathcal{T}_{2,u}] \le \frac{L_u \gamma_u^2 \tau_u^2 \sigma_u^2}{m} + L_u \gamma_u^2 \tau_u \sum_{k=0}^{\tau_u - 1} \mathbf{E}_t \underbrace{\left\| \frac{1}{m} \sum_{i \in S^{(t)}} \nabla_u F_i \left( \tilde{u}_{i,k}^{(t)}, \tilde{v}_i^{(t+1)} \right) \right\|^2}_{=:\mathcal{T}_k'}.$$

We break the term $\mathcal{T}_k'$ as

$$\mathcal{T}_k' \le 3 \left\| \frac{1}{m} \sum_{i \in S^{(t)}} \left( \nabla_u F_i \left( \tilde{u}_{i,k}^{(t)}, \tilde{v}_i^{(t+1)} \right) - \nabla_u F_i \left( u^{(t)}, \tilde{v}_i^{(t+1)} \right) \right) \right\|^2$$

$$+ 3 \left\| \frac{1}{m} \sum_{i \in S^{(t)}} \nabla_u F_i \left( u^{(t)}, \tilde{v}_i^{(t+1)} \right) - \nabla_u F \left( u^{(t)}, \widetilde{V}^{(t+1)} \right) \right\|^2 + 3 \left\| \nabla_u F \left( u^{(t)}, \widetilde{V}^{(t+1)} \right) \right\|^2.$$

For the first term, we use Jensen's inequality to take the squared norm inside the sum, then use smoothness and take an expectation over the sampling of devices to get

$$\mathbf{E}_t \left\| \frac{1}{m} \sum_{i \in S^{(t)}} \left( \nabla_u F_i \left( \tilde{u}_{i,k}^{(t)}, \tilde{v}_i^{(t+1)} \right) - \nabla_u F_i \left( u^{(t)}, \tilde{v}_i^{(t+1)} \right) \right) \right\|^2 \le \frac{L_u^2}{n} \sum_{i=1}^{n} \mathbf{E}_t \left\| \tilde{u}_{i,k}^{(t)} - u^{(t)} \right\|^2.$$

For the second term, we use the fact that $S^{(t)}$ was sampled without replacement (cf. Lemma 17) and invoke the gradient diversity assumption (Assumption 3') to get,

$$\left\| \frac{1}{m} \sum_{i \in S^{(t)}} \nabla_u F_i \left( u^{(t)}, \tilde{v}_i^{(t+1)} \right) - \nabla_u F \left( u^{(t)}, \widetilde{V}^{(t+1)} \right) \right\|^2$$

$$\le \left( \frac{n-m}{n-1} \right) \frac{1}{mn} \sum_{i=1}^{n} \left\| \nabla_u F_i \left( u^{(t)}, \tilde{v}_i^{(t+1)} \right) - \nabla_u F \left( u, \widetilde{V}^{(t+1)} \right) \right\|^2$$

$$\le \frac{2}{m} \left( 1 - \frac{m}{n} \right) \left( \delta^2 + \rho^2 \mathbf{E}_t \left\| \nabla_u F \left( u^{(t)}, \widetilde{V}^{(t+1)} \right) \right\|^2 \right).$$

To complete the proof, we plug these terms back into the definition of $\mathcal{T}_k'$ and $\mathbf{E}_t[\mathcal{T}_{2,u}]$ to complete the proof. $\qquad\square$

**Claim 13** (Bounding $\mathcal{T}_{3,u}$). *We have,*

$$\mathbf{E}_t [\mathcal{T}_{3,u}] \le 8 \gamma_v^2 \tau_v^2 L_v \chi^2 \left( 1 - \frac{m}{n} \right) \Delta_v^{(t)} + 4 \chi^2 \gamma_v^2 \tau_v^2 L_v \sigma_v^2 \left( 1 - \frac{m}{n} \right).$$

*Proof.* Since $v_i^{(t+1)} = \tilde{v}_i^{(t+1)}$ for $i \in S^{(t)}$, we have that

$$\mathcal{T}_{3,u} = \frac{\chi^2 L_v}{2n} \sum_{i \notin S^{(t)}} \left\| \tilde{v}_i^{(t+1)} - v_i^{(t)} \right\|^2.$$

Since $\left\| \tilde{v}_i^{(t+1)} - v_i^{(t)} \right\|^2$ is independent of $S^{(t)}$, we can take an expectation to get

$$\mathbf{E}_t[\mathcal{T}_{3,u}] = \frac{\chi^2 L_v}{2n} \sum_{i=1}^{n} \mathbb{P}(i \notin S^{(t)}) \mathbf{E}_t \left\| \tilde{v}_i^{(t+1)} - v_i^{(t)} \right\|^2$$

$$= \frac{\chi^2 L_v}{2n} \left( 1 - \frac{m}{n} \right) \sum_{i=1}^{n} \mathbf{E}_t \left\| \tilde{v}_i^{(t+1)} - v_i^{(t)} \right\|^2.$$

Plugging in Lemma 19 completes the proof. $\qquad\square$

The analysis of the $v$-step is given in the next result.

**Claim 14.** *Consider the setting of Theorem 2 and assume that $\gamma_v \tau_v L_v \leq 1/8$. We have,*

$$\mathbf{E}_t\left[F\left(u^{(t)}, V^{(t+1)}\right) - F\left(u^{(t)}, V^{(t)}\right)\right] \leq -\frac{\gamma_v \tau_v m \Delta_v^{(t)}}{8n} + \frac{\gamma_v^2 \tau_v^2 L_v \sigma_v^2 m}{2n} + \frac{4\gamma_v^3 L_v^2 \tau_v^2 (\tau_v - 1)\sigma_v^2 m}{n}.$$

*Proof.* From smoothness, we get,

$$F_i\left(u^{(t)}, \tilde{v}_i^{(t+1)}\right) - F_i\left(u^{(t)}, v_i^{(t)}\right) \leq \underbrace{\left\langle \nabla_v F_i\left(u^{(t)}, v_i^{(t)}\right), \tilde{v}_i^{(t+1)} - v_i^{(t)}\right\rangle}_{\mathcal{T}_{1,v}} + \underbrace{\frac{L_v}{2}\left\|\tilde{v}_i^{(t+1)} - v_i^{(t)}\right\|^2}_{\mathcal{T}_{2,v}}.$$

We bound the first term as

$$\mathbf{E}_t[\mathcal{T}_{1,v}] = -\gamma_v \mathbf{E}_t\left\langle \nabla_v F_i\left(u^{(t)}, v_i^{(t)}\right), \sum_{k=0}^{\tau_v - 1} \nabla_v F_i\left(u^{(t)}, \tilde{v}_{i,k}^{(t)}\right)\right\rangle$$

$$= -\gamma_v \tau_v \left\|\nabla_v F_i\left(u^{(t)}, v_i^{(t)}\right)\right\|^2$$

$$- \gamma_v \sum_{k=0}^{\tau_v - 1} \mathbf{E}_t\left\langle \nabla_v F_i\left(u^{(t)}, v_i^{(t)}\right), \nabla_v F_i\left(u^{(t)}, \tilde{v}_{i,k}^{(t)}\right) - \nabla_v F_i\left(u^{(t)}, v_i^{(t)}\right)\right\rangle$$

$$\leq -\frac{\gamma_v \tau_v}{2}\left\|\nabla_v F_i\left(u^{(t)}, v_i^{(t)}\right)\right\|^2 + \frac{\gamma_v}{2} \sum_{k=0}^{\tau_v - 1} \mathbf{E}_t\left\|\nabla_v F_i\left(u^{(t)}, \tilde{v}_{i,k}^{(t)}\right) - \nabla_v F_i\left(u^{(t)}, v_i^{(t)}\right)\right\|^2$$

$$\leq -\frac{\gamma_v \tau_v}{2}\left\|\nabla_v F_i\left(u^{(t)}, v_i^{(t)}\right)\right\|^2 + \frac{\gamma_v L_v^2}{2} \sum_{k=0}^{\tau_v - 1}\left\|\tilde{v}_{i,k}^{(t)} - v_i^{(t)}\right\|^2.$$

Next, we observe that

$$\mathbf{E}_z\|G_{i,v}(u, v_i, z)\|^2 = \|\nabla_v F_i(u, v_i)\|^2 + \mathbf{E}_z\|G_{i,v}(u, v_i, z) - \nabla_v F_i(u, v_i)\|^2 \leq \|\nabla_v F_i(u, v_i)\|^2 + \sigma_v^2.$$

We invoke this inequality to handle the second term as

$$\mathbf{E}_t[\mathcal{T}_{2,v}] \leq \frac{\gamma_v^2 L_v \tau_v}{2} \sum_{k=0}^{\tau_v - 1} \mathbf{E}_t\left\|G_{i,v}\left(u^{(t)}, \tilde{v}_{i,k}^{(t)}, z_{i,k}^{(t)}\right)\right\|^2$$

$$\leq \frac{\gamma_v^2 L_v \tau_v^2 \sigma_v^2}{2} + \frac{\gamma_v^2 L_v \tau_v}{2} \sum_{k=0}^{\tau_v - 1} \mathbf{E}_t\left\|\nabla_v F_i\left(u^{(t)}, \tilde{v}_{i,k}^{(t)}\right)\right\|^2$$

$$\leq \frac{\gamma_v^2 L_v \tau_v^2 \sigma_v^2}{2} + \gamma_v^2 L_v \tau_v^2 \left\|\nabla_v F_i\left(u^{(t)}, v_i^{(t)}\right)\right\|^2$$

$$+ \gamma_v^2 L_v \tau_v \sum_{k=0}^{\tau_v - 1} \mathbf{E}_t\left\|\nabla_v F_i\left(u^{(t)}, \tilde{v}_{i,k}^{(t)}\right) - \nabla_v F_i\left(u^{(t)}, v_i^{(t)}\right)\right\|^2$$

$$\leq \frac{\gamma_v^2 L_v \tau_v^2 \sigma_v^2}{2} + \gamma_v^2 L_v \tau_v^2\left\|\nabla_v F_i\left(u^{(t)}, v_i^{(t)}\right)\right\|^2 + \gamma_v^2 L_v^3 \tau_v \sum_{k=0}^{\tau_v - 1} \mathbf{E}_t\left\|\tilde{v}_{i,k}^{(t)} - v_i^{(t)}\right\|^2.$$

Plugging these bounds for $\mathcal{T}_{1,v}$ and $\mathcal{T}_{2,v}$ into the initial smoothness bound and using $\gamma_v L_v \tau_v \leq 1/4$ gives

$$\mathbf{E}_t\left[F_i\left(u^{(t)}, \tilde{v}_i^{(t+1)}\right) - F_i\left(u^{(t)}, v_i^{(t)}\right)\right] \leq$$

$$-\frac{\gamma_v \tau_v}{4}\left\|\nabla_v F_i\left(u^{(t)}, v_i^{(t)}\right)\right\|^2 + \gamma_v L_v^2 \sum_{k=0}^{\tau_v - 1}\left\|\tilde{v}_{i,k}^{(t)} - v_i^{(t)}\right\|^2 + \frac{\gamma_v^2 L_v \tau_v^2 \sigma_v^2}{2}.$$

We invoke Lemma 18 to bound the $\sum_k \mathbf{E}_t\|\tilde{v}_{i,k}^{(t)} - v_i^{(t)}\|^2$ term, which is also known as client drift. We simplify some coefficients using $8\gamma_v \tau_v L_v \leq 1$ to get

$$\mathbf{E}_t\left[F_i\left(u^{(t)}, \tilde{v}_i^{(t+1)}\right) - F_i\left(u^{(t)}, v_i^{(t)}\right)\right] \leq$$

$$-\frac{\gamma_v \tau_v}{8}\left\|\nabla_v F_i\left(u^{(t)}, v_i^{(t)}\right)\right\|^2 + \frac{\gamma_v^2 L_v \tau_v^2 \sigma_v^2}{2} + 4\gamma_v^3 L_v \tau_v^2 (\tau_v - 1)\sigma_v^2.$$

It remains to invoke that $S^{(t)}$ is a uniformly random sample of $m$ devices from $\{1, \cdots, n\}$ and that $\tilde{v}_i^{(t+1)}$ is independent of $S^{(t)}$. To this end, note that

$$\mathbf{E}_t \left[ F \left( u^{(t)}, V^{(t+1)} \right) - F \left( u^{(t)}, V^{(t)} \right) \right] = \frac{m}{n} \mathbf{E}_t \left[ \frac{1}{m} \sum_{i \in S^{(t)}} F_i \left( u^{(t)}, \tilde{v}_i^{(t+1)} \right) - F_i \left( u^{(t)}, v_i^{(t)} \right) \right]$$

$$\leq \frac{m}{n^2} \sum_{i=1}^{n} \mathbf{E}_t \left[ F_i \left( u^{(t)}, \tilde{v}_i^{(t+1)} \right) - F_i \left( u^{(t)}, v_i^{(t)} \right) \right] .$$

Plugging in the previous bound completes the proof. $\qquad \square$

**Remark 15.** *We only invoked the partial gradient diversity assumption (Assumption 3) at (virtual) iterates $(u^{(t)}, \widetilde{V}^{(t+1)})$; therefore, it suffices if the assumption only holds at iterates $(u^{(t)}, \widetilde{V}^{(t+1)})$ generated by FedAlt, rather than at all $(u, V)$.*

### A.4 TECHNICAL LEMMAS

The first lemma involves smoothness of two blocks of variables; we use this in the proof of FedSim.

**Lemma 16** (Block Smoothness). *Suppose $F_i : \mathbb{R}^d \times \mathbb{R}^{d_i}$ satisfy Assumption 1'. Then, it holds that*

$$F_i(w', v_i') - F_i(w, v_i) \leq \langle \nabla_w F_i(w, v_i), w' - w \rangle + \langle \nabla_v F_i(w, v_i), v_i' - v_i \rangle$$

$$+ \frac{L_w}{2}(1 + \chi^2)\|w' - w\|^2 + \frac{L_v}{2}(1 + \chi^2)\|v_i' - v_i\|^2 .$$

*Proof.* Using the $L_w$-smoothness of $F(\cdot, v_i')$ and the $L_v$-smoothness of $F(w, \cdot)$, we have

$$F_i(w', v_i') - F_i(w, v_i') \leq \langle \nabla_w F_i(w, v_i'), w' - w \rangle + \frac{L_w}{2}\|w' - w\|^2,$$

$$F_i(w, v_i') - F_i(w, v_i) \leq \langle \nabla_w F_i(w, v_i), v_i' - v_i \rangle + \frac{L_v}{2}\|v_i' - v_i\|^2.$$

Summing the above two inequalities together gives

$$F_i(w', v_i') - F_i(w, v_i) \leq \langle \nabla_w F_i(w, v_i'), w' - w \rangle + \langle \nabla_v F_i(w, v_i), v_i' - v_i \rangle$$

$$+ \frac{L_w}{2}\|w' - w\|^2 + \frac{L_v}{2}\|v_i' - v_i\|^2 . \tag{16}$$

We can bound the first inner product term on the right-hand side of the above inequality as

$$\langle \nabla_w F_i(w, v_i'), w' - w \rangle = \langle \nabla_w F_i(w, v_i), w' - w \rangle + \langle \nabla_w F_i(w, v_i') - \nabla_w F_i(w, v_i), w' - w \rangle$$

$$\leq \langle \nabla_w F_i(w, v_i), w' - w \rangle + \|\nabla_w F_i(w, v_i') - \nabla_w F_i(w, v_i)\|\|w' - w\|$$

$$\leq \langle \nabla_w F_i(w, v_i), w' - w \rangle + L_{wv}\|v_i' - v_i\|\|w' - w\|$$

$$\leq \langle \nabla_w F_i(w, v_i), w' - w \rangle + \chi\sqrt{L_w L_v}\|v_i' - v_i\|\|w' - w\|$$

$$\leq \langle \nabla_w F_i(w, v_i), w' - w \rangle + \chi^2 \frac{L_v}{2}\|v_i' - v_i\|^2 + \chi^2 \frac{L_w}{2}\|w' - w\|^2,$$

where the first inequality is due to Cauchy-Schwarz, the second inequality is due to $L_{wv}$-Lipschitz property of $\nabla_w F_i(w, \cdot)$, the third inequality is due to the definition of $\chi$ in (5), and the last inequality is due to Young's inequality. Substituting the above inequality into (16) yields the desired result. $\quad \square$

Next, we have the variance of sampling without replacement. Note the correction factor of $(n - m)/(n - 1)$ over sampling with replacement. We include the elementary proof for completeness.

**Lemma 17** (Sampling Without Replacement). *Let $a_1, \cdots, a_n \in \mathbb{R}^d$ be given. Let $S$ be a uniformly random sample of size $m$ from this collection, where the sampling is without replacement. Denoting the mean $\bar{a} = \sum_{i=1}^{n} a_i/n$, we have,*

$$\mathbf{E}_S \left\| \frac{1}{m} \sum_{i \in S} a_i - \bar{a} \right\|^2 \leq \left( \frac{n - m}{n - 1} \right) \frac{1}{m} \left( \frac{1}{n} \sum_{i=1}^{n} \|a_i - \bar{a}\|^2 \right) .$$

*Proof.* The statement is trivially true if $m = 1$ or $m = n$. Therefore, we assume now that $2 \leq m \leq n - 1$. Further, without loss of generality, we assume that $\bar{a} = 0$. Finally, let $\mathcal{S}$ denote the set of all subsets of $[n]$ of size $m$. Note that $|\mathcal{S}| = \binom{n}{m}$. We now have,

$$\mathbf{E}_S \left\| \frac{1}{m} \sum_{i \in S} a_i \right\|^2 = \frac{1}{m^2 \binom{n}{m}} \sum_{S \in \mathcal{S}} \left( \sum_{i \in S} \|a_i\|^2 + \sum_{i,j \in S : i \neq j} \langle a_i, a_j \rangle \right) .$$

For the first term, we have,

$$\sum_{S \in \mathcal{S}} \sum_{i \in S} \|a_i\|^2 = \sum_{i=1}^n \sum_{S \in \mathcal{S} : i \in S} \|a_i\|^2 = \binom{n-1}{m-1} \sum_{i=1}^n \|a_i\|^2 .$$

Likewise, for the second term, we use $\sum_{j \neq i} a_j = -a_i$ to get,

$$\sum_{i,j \in S : i \neq j} \langle a_i, a_j \rangle = \sum_{i=1}^n \sum_{j \neq i} \sum_{S \in \mathcal{S} : i,j \in S} \langle a_i, a_j \rangle = \binom{n-2}{m-2} \sum_{i=1}^n \sum_{j \neq i} \langle a_i, a_j \rangle = -\binom{n-2}{m-2} \sum_{i=1}^n \|a_i\|^2 .$$

Therefore, we get,

$$\mathbf{E}_S \left\| \frac{1}{m} \sum_{i \in S} a_i \right\|^2 = \frac{\binom{n-1}{m-1} - \binom{n-2}{m-2}}{m^2 \binom{n}{m}} \sum_{i=1}^n \|a_i\|^2 = \frac{\binom{n-2}{m-1}}{m^2 \binom{n}{m}} \sum_{i=1}^n \|a_i\|^2 = \frac{n-m}{mn(n-1)} \sum_{i=1}^n \|a_i\|^2 .$$

$\square$

The next two lemmas are about the effect of the local updates in the local SGD literature. The first lemma has also appeared in (Karimireddy et al., 2020); we give the proof for completeness.

**Lemma 18.** *Consider $f : \mathbb{R}^d \to \mathbb{R}$ which is L-smooth and fix a $w^{(0)} \in \mathbb{R}^d$. Define the sequence $(w^{(t)})$ of iterates produced by stochastic gradient descent with a fixed learning rate $\gamma$ starting from $w^{(0)}$:*

$$w^{(t+1)} = w^{(t)} - \gamma g^{(t)} ,$$

*where $g^{(t)}$ is an unbiased (and independent of $w$) estimator of $\nabla f(w)$ with bounded variance $\sigma^2$. Fix a number $\tau$ of steps. If $\gamma \leq (\sqrt{2}\tau L)^{-1}$, we have the bound*

$$\sum_{t=0}^{\tau-1} \|w^{(t)} - w^{(0)}\|^2 \leq 8\gamma^2 \tau^2 (\tau - 1) \|\nabla f(w^{(0)})\|^2 + 4\gamma^2 \tau^2 (\tau - 1)\sigma^2 .$$

*Proof.* If $\tau = 1$, we have nothing to prove. Assume now that $\tau \geq 2$. Let $\mathcal{F}^{(t)}$ be the sigma-algebra generated by $w^{(t)}$ and denote $\mathbf{E}_t[\cdot] = \mathbf{E}[\cdot | \mathcal{F}^{(t)}]$. We will use the inequality

$$\mathbf{E}_t \left\| g^{(t)} \right\|^2 = \mathbf{E}_t \left\| g^{(t)} - \nabla f(w^{(t)}) \right\|^2 + \left\| \nabla f(w^{(t)}) \right\|^2 \leq \sigma^2 + \left\| \nabla f(w^{(t)}) \right\|^2 . \qquad (17)$$

We now successively deduce,

$\mathbf{E}_t \|w^{(t+1)} - w^{(0)}\|^2 = \|w^{(t)} - w^{(0)} - \gamma g^{(t)}\|^2$

$\overset{(a)}{\leq} \left(1 + \frac{1}{\tau - 1}\right) \|w^{(t)} - w^{(0)}\|^2 + \gamma^2 \tau \mathbf{E}_t \|g^{(t)}\|^2$

$\overset{(b)}{\leq} \left(1 + \frac{1}{\tau - 1}\right) \|w^{(t)} - w^{(0)}\|^2 + 2\gamma^2 \tau \|\nabla f(w^{(t)}) - \nabla f(w^{(0)})\|^2 + 2\gamma^2 \tau \|\nabla f(w^{(0)})\|^2 + \gamma^2 \tau \sigma^2$

$\overset{(c)}{\leq} \left(1 + \frac{1}{\tau - 1} + 2\gamma^2 \tau L^2\right) \|w^{(t)} - w^{(0)}\|^2 + 2\gamma^2 \tau \|\nabla f(w^{(0)})\|^2 + \gamma^2 \tau \sigma^2$

$\overset{(d)}{\leq} \left(1 + \frac{2}{\tau - 1}\right) \|w^{(t)} - w^{(0)}\|^2 + 2\gamma^2 \tau \|\nabla f(w^{(0)})\|^2 + \gamma^2 \tau \sigma^2 .$

Above, we used (a) the inequality $2\alpha\beta \leq \alpha^2/\delta^2 + \delta^2\beta^2$ for reals $\alpha, \beta, \delta$, (b) Eq. (17), (c) $L$-smoothness of $f$, and, (d) the condition on the learning rate.

Let $C = 2\gamma^2\tau\|\nabla f(w^{(0)})\|^2 + \gamma^2\tau\sigma^2$. Unrolling the inequality and summing up the series gives for all $t \leq \tau - 1$

$$\|w^{(t)} - w^{(0)}\|^2 \leq C\sum_{j=0}^{t-1}\left(1 + \frac{2}{\tau-1}\right)^j \leq \frac{C}{2}(\tau-1)\left(1 + \frac{2}{\tau-1}\right)^t$$

$$\leq \frac{C}{2}(\tau-1)\left(1 + \frac{2}{\tau-1}\right)^{\tau-1} \leq \frac{C}{2}(\tau-1)e^2,$$

where we used the bound $(1 + 1/\alpha)^\alpha \leq e$ for all $\alpha > 0$. Summing over $t$ and using the numerical bound $e^2 < 8$ completes the proof. $\square$

**Lemma 19.** *Consider the setting of Lemma 18. If $\gamma \leq (2\tau L)^{-1}$, we have the bound*

$$\|w^{(\tau)} - w^{(0)}\|^2 \leq 16\gamma^2\tau^2\|\nabla f(w^{(0)})\|^2 + 8\gamma^2\tau^2\sigma^2.$$

*Proof.* Proceeding similar to the last proof (expect using $\delta = \tau$) gives us

$$\mathbf{E}_t\left\|w^{(t+1)} - w^{(0)}\right\|^2 \leq \left(1 + \frac{2}{\tau}\right)\left\|w^{(t)} - w^{(0)}\right\|^2 + 4\gamma^2\tau\left\|\nabla f(w^{(0)})\right\|^2 + 2\gamma^2\tau\sigma^2.$$

Unrolling and summing up the sequence completes the proof, similar to that of Lemma 18. $\square$

Finally, the last lemma is about constants.

**Lemma 20.** *Let $\gamma_u, \gamma_v, L_w, L_v, \chi, f \in \mathbb{R}_+$ and a natural number $\tau$ be given. Denote*

$$A := \gamma_u L_u^2 + f\gamma_v\chi^2 L_u L_v, \quad and, \quad B := f\gamma_v L_v^2 + \gamma_u\chi^2 L_u L_v.$$

*Suppose $\gamma_u = c_u/(\tau L_u)$ and $\gamma_v = c_v/(\tau L_v)$ with $c_u, c_v > 0$ satisfying*

$$c_u, c_v \leq \frac{1}{\sqrt{6}}\max\{1, \chi^{-2}\}.$$

*Then, we have that*

$$\gamma_v^2\chi^2 L_u L_v B + \gamma_u^2 L_u^2 A \leq A/(3\tau^2), \quad and, \quad \gamma_u^2\chi^2 L_u L_v A + \gamma_v^2 L_v^2 B \leq B/(3\tau^2).$$

*Proof.* Note that it suffices to show

$$3\tau^2\chi^2\gamma_v^2 L_u L_v B \leq A/2, \quad and, \quad 3\tau^2\chi^2\gamma_u^2 L_u L_v A \leq B/2.$$

Plugging in $\gamma_u, \gamma_v$, these are equivalent to

$$6\chi^2 fc_v^3 + 6\chi^4 c_v^2 c_u \leq \chi^2 fc_v + c_u \quad and, \quad 6\chi^2 c_u^3 + 6\chi^4 fc_v c_u^2 \leq fc_v + \chi^2 c_u.$$

The assumption on $c_v$ implies that $6\chi^2 fc_v^3 \leq \chi^2 fc_v$ and $6\chi^4 c_v^2 c_u \leq c_u$. Therefore, the first condition holds. Similarly, the second condition holds too. $\square$

# B EXPERIMENTS: DETAILED SETUP AND HYPERPARAMETERS

We conduct our experiments on three datasets from two modalities, namely images and text. The datasets contain a natural, non-iid split of data which is reflective of data heterogeneity encountered in federated learning. We describe in detail the experimental setup and hyperparameters. The code to reproduce the experimental results will be publicly released.

The outline of this section is:

- §B.1 describes the tasks and their associated datasets and metrics.
- §B.2 describes the experimental pipeline as well as the baselines we compare to.
- §B.3 presents the hyperparameters of all the algorithms.

As discussed in §1, we take the weight $\alpha_k$ to be proportional to the number of datapoints available on the device.

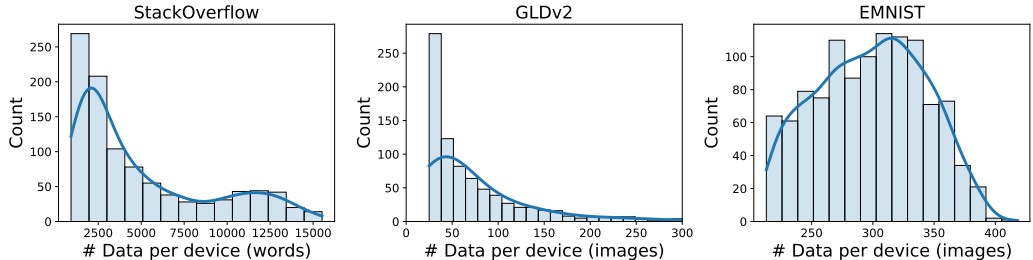

Figure 5: Distribution of number of training samples per device for each of the tasks considered in the experiments. For GLDv2, we do not show the long right tail, where the maximum number of data points per device is 1000 (cf. Table 2).

### B.1 DATASETS, TASKS AND MODELS

We consider three tasks motivated by real-world applications of federated learning. The tasks are summarized in Table 2 of the main paper and the distribution of data across the clients is visualized in Figure 5.

For each model, we consider three partial personalization architectures:

(a) **Input layer personalization**: Motivated by Liang et al. (2019), this architecture places the first layer on-device to learn a personalized representation per-client, while the rest of the model is shared. For the transformer model, we use the first transformer layer in place of the word embedding layer owing to its large size.

(b) **Output layer personalization**: Motivated by Collins et al. (2021), this architecture learns a shared global representation but personalizes the prediction layer. For the transformer model, we use the last transformer layer in place of the last prediction layer owing to its large size.

(c) **Adapter personalization**: We also consider a novel partial personalization architecture, where the full model is shared among all clients, while each client adds personalized adapter modules, which are lightweight modules added between layers of the shared model. We use the transformer adapters proposed by Houlsby et al. (2019) and residual adapters proposed by Rebuffi et al. (2017).

#### B.1.1 STACKOVERFLOW FOR NEXT WORD PREDICTION

**Dataset.** The StackOverflow dataset comprises of questions and answers from the programming question-answer website stackoverflow.com. The goal of the next word prediction task is to predict the next word given a partial sequence of words in a question or answer. This task is a good open-source benchmark for next word predictions in mobile keyboards. We use the StackOverflow dataset provided by TensorFlow Federated.

**Client Distributions.** Each client corresponds to one user on Stack Overflow; the data on the client corresponds to the questions and answers posted by this user. We only consider clients with at least 100 training sequences and 10 testing sequences, where a sequence refers to either a question or an answer. We use a fixed subsample of 1000 of them. Following Reddi et al. (2021), we restrict the vocabulary to the top 10000 most frequently occurring words in the dataset. We pad and truncate each sequence of each client to length 20 and consider at most 1000 training sequences on each client.

**Model.** We use a transformer model (Vaswani et al., 2017) commensurate in size with BERT Mini (Turc et al., 2019). It has with 4 transformer blocks and 4 attention heads in each self-attention layer with a transformer hidden dimension of 256 and a fully-connected hidden dimension of 1024. The output layer is a causal language modeling head, i.e., a fully connected layer which assigns

Table 5: Summary of partial personalization architectures for the transformer model for next word prediction.

| Personalization Type | Layer on-device | # Personalized Params. | # Shared Params. |
|---|---|---|---|
| Input Layer | 1st transformer block | $0.8M$ | $4.9M$ |
| Output Layer | Last transformer block | $0.8M$ | $4.9M$ |
| Adapter | Adapter modules | $0.07M$ | $5.7M$ |

Table 6: Summary of partial personalization architectures for the ResNet-18 model for visual landmark recognition.

| Personalization Type | Layer on-device | # Personalized Params. | # Shared Params. |
|---|---|---|---|
| Input Layer | 1st conv. layer | $0.01M$ | $12.2M$ |
| Output Layer | Last fully connected layer | $1M$ | $11.2M$ |
| Adapter | Residual adapter modules | $1.4M$ | $12.2M$ |

a score for each possible vocabulary item, including the special tokens. The model has 6 million parameters, which require around 23 megabytes of memory.

**Partial Personalization Architecture.** The partial personalization architectures used are summarized in Table 5.

**Loss Function and Evaluation Metric.** We train the model with the causal language modeling objective. That is, for each partial sequence, we treat the prediction of the next word as a multiclass classification problem to minimize the multinomial logistic loss, also known as cross entropy loss. For evaluation, we use the top-1 accuracy of predicting words in the proper 10000-word vocabulary (i.e., ignoring special tokens such as padding, out-of-vocabulary, and beginning/end of sequence).

### B.1.2 GLDv2 FOR VISUAL LANDMARK RECOGNITION

**Dataset.** GLDv2 stands for Google Landmarks Dataset v2 (Weyand et al., 2020), which is a large-scale image dataset. It contains images of popular landmarks from around the world taken and uploaded by Wikipedia contributors. While the images vary in size, the most common image size is $800 \times 600$ pixels.

The goal of the visual landmark recognition task is to identify the landmark from its image. This task resembles a scenario where smartphone users take photos of natural and architectural landmarks while traveling. We use the federated version of the GLDv2 dataset introduced by Hsu et al. (2020) with 2028 landmarks and provided by TensorFlow Federated.

**Client Distributions.** Each client corresponds to one Wikipedia user and contains all the images contributed by that user. We only all 823 clients with at least 50 datapoints. We do not use original test set from GLDv2 from evaluation as it comes from different clients. Instead, we take 50% of the data on each client as a testing set.

**Model.** We use a ResNet-18 (He et al., 2016) model pretrained on ImageNet (Deng et al., 2009), with group normalization instead of batch normalization (Hsieh et al., 2020). We resize all images to $224 \times 224$. We use two data augmentations for training: a random crop from $256 \times 256$ and a random horizontal flip. The model has 12 million parameters, which require around 49 megabytes of storage.

**Partial Personalization Architecture.** The partial personalization architectures used are summarized in Table 6.

**Loss Function and Evaluation Metric.** We use the multinomial logistic loss, also known as cross entropy loss. We evaluate the performance of the model using its classification accuracy.

Table 7: Summary of partial personalization architectures for the ResNet-18 model for character recognition.

| Personalization Type | Layer on-device | # Personalized Params. | # Shared Params. |
|---|---|---|---|
| Input Layer | 1st conv. layer | $0.7K$ | $11.2M$ |
| Output Layer | Last fully connected layer | $0.03M$ | $11.2M$ |
| Adapter | Residual adapter modules | $1.4M$ | $11.2M$ |

### B.1.3 EMNIST FOR CHARACTER RECOGNITION

**Dataset.** EMNIST (Cohen et al., 2017) is a character recognition dataset. The goal is to identify images of handwritten digits or letters; there are 62 possible options (a-z,A-Z, 0-9). The images are grey-scaled pictures of $28 \times 28 = 784$ pixels. We use the EMNIST dataset provided by TensorFlow Federated.

**Client Distributions.** Each client corresponds to one "writer", i.e., the human subject who hand-wrote the digit/letter during the data collection process. We only use those clients with at least 100 training points and 25 testing points: there are $1114$ of such clients.

**Model.** We use a ResNet-18 (He et al., 2016) model with group normalization instead of batch normalization (Hsieh et al., 2020). We make two modifications to handle the smaller image size ($28 \times 28 \times 1$ as opposed to the $224 \times 224 \times 3$ which the original ResNet was designed to accept): (a) we use a convolutional kernel of size $3 \times 3$ rather than the original $7 \times 7$ in the first convolution layer, and, (b) we drop the first pooling layer. The model has 11 million parameters, which require around 45 megabytes. Note that the number of parameters in this ResNet is smaller than the one for GLDv2 due to the architectural modifications we make for smaller images as well as the smaller number of classes.

**Partial Personalization Architecture.** The partial personalization architectures used are summarized in Table 7.

**Loss Function and Evaluation Metric.** We use the multinomial logistic loss, also known as cross entropy loss. We evaluate the performance of the model using its classification accuracy.

### B.2 EXPERIMENTAL PIPELINE AND BASELINES

There are three components in the training pipeline for all experiments:

(a) Non-personalized federated training: The first step involves training a global model $w_g$ using the one-model-fits-all approach of (1) with FedAvg variants.

(b) Personalized federated training: This optional second step involves training the shared parameters $w$ together with the personalized parameters $v_k$ using a personalized federated learning approach. We warm-start $w, v_k$ from the non-personalized model $w_g$ from the previous step.

(c) Final finetuning: The last step involves only finetuning the personalized parameters $v_k$ while the shared parameters $w$ remain unchanged.

For step (b), we initialize $v_k$ for each $k$ to be the appropriate part of $w_g$ for input/output layer personalization. On the other hand, for adapters, we initialize $v_k$ to be equal to the *same* set of randomly initialized weights for each device $k$.

We consider the following baselines:

- **Non-personalized**: This denotes the performance of step (a) of the pipeline above, i.e., non-personalized federated training with FedAvg variants.

- **Full model personalization**: We consider three baselines of personalization of the full model:

  (i) **Finetune**: The non-personalized model from step (a) of the pipeline above is finetuned locally on each client (step (c) of the pipeline). Step (b) is skipped for this baseline.

Table 8: Hyperparameters for each dataset/task.

| | Hyperparameter | StackOverflow | GLDv2 | EMNIST |
|---|---|---|---|---|
| | Batch size | 64 | 64 | 32 |
| | Devices per round | 50 | 50 | 10 |
| | Local epochs | 1 | 1 | 1 |
| | Server Optimizer | FedAdam | FedAdam | FedAvg |
| Common | Client Optimizer | SGD | SGD | SGD |
| | Global Scheduler | Linear | Linear | Exponential |
| | Warm up | 10% of rounds | 10% of rounds | N/A |
| | LR decay rounds | N/A | N/A | 500 |
| | Max. grad. norm. | 0.1 | N/A | N/A |
| Non-personalized training (step (a) of the pipeline) | # Rounds | 1000 | 2500 | 2000 |
| | Server learning rate | $5 \times 10^{-4}$ | $2 \times 10^{-4}$ | 1.0 |
| | Client learning rate | 1 | $10^{-2}$ | 0.5 |
| Personalized training (step (b) of the pipeline) | # Rounds | 500 | 600 | 500 |
| | Server learning rate | $5 \times 10^{-5}$ | $2 \times 10^{-5}$ | 1.0 |
| | Client learning rate | $10^{-1}$ | $10^{-3}$ | $10^{-2}$ |
| Local finetuning (step (c) of the pipeline) | #Epochs | 5 | 5 | 5 |
| | Optimizer | SGD | SGD | SGD |
| | Client learning rate | $10^{-1}$ | $10^{-3}$ | $10^{-2}$ |

(ii) **Ditto** (Li et al., 2021): The non-personalized model from step (a) of the pipeline above is finetuned locally on each client (step (c) of the pipeline) with $\ell_2$ regularization $\|v - w_g\|^2$. Step (b) is skipped for this baseline.

(iii) **pFedMe** (Dinh et al., 2020): The non-personalized baseline model from step (a) is trained further in step (b) to optimize (2) using the pFedMe algorithm of Dinh et al. (2020). Finally the resulting model $w$ is finetuned locally in step (c).

- **Partial Model Personalization**: We consider partial model personalization with three different architectures, as defined in §B.1. For each personalization approach, we start with the non-personalized model in step (a), continue personalization in step (b) using either FedAlt or FedSim as the algorithm, and finally run step (c) for the local finetuning.

### B.3 HYPERPARAMETERS AND EVALUATION DETAILS

All the tuning of hyperparameters was performed on validation data, formed by holding out 20% of the training data on each device. Once the tuning was complete, we reran the experiments on the full training data, including those held out for validation.

**Evaluation Metric.** Our primary evaluation metric is the weighted average of the test accuracy on each client, weighted by the number of test examples (the details of how the accuracy is computed on each dataset is given in §B.1 in the paragraph on "Loss Function and Evaluation Metric"). This corresponds to the unweighted accuracy obtained by pooling all the data locally, similar to the loss as discussed in §1. The same metric is used for hyperparameter tuning and is reported in all the tables and plots, unless explicitly noted otherwise.

The final hyperparameters we use are given in Table 8.

**Rounds.** We start with the number of communication rounds (i.e., the number of calls to secure aggregation routine for the shared parameters), which is used to measure the progress of each algorithm. For the non-personalized training, we use 1000 rounds for StackOverflow, 2500 rounds for GLDv2 and 2000 rounds for EMNIST. For the personalized training, we warm-start the model from the non-personalized one, and run the training for 500 rounds for StackOverflow and EMNIST and 600 rounds for GLDv2.

**Devices per Round.** All devices are assumed to be available and selections are made uniformly at random. Following (Reddi et al., 2021; Weyand et al., 2020), we select 50 devices per round for

Table 9: This table shows Table 4 of the main paper along with the corresponding standard deviations as subscripts. We compare FedAlt and FedSim for partial model personalization in this table. "FT (part.)" corresponds to finetuning the personal parameters $v_i$ locally while fixing the shared parameters $u$ from a non-personalized training. The numbers are averaged over 5 random seeds; the boldfaced numbers denote those within 1 standard deviation of the highest accuracy in each row.

| | StackOverflow | | | GLDv2 | | | EMNIST | | |
|---|---|---|---|---|---|---|---|---|---|
| | FT (part.) | FedAlt | FedSim | FT (part.) | FedAlt | FedSim | FT (part.) | FedAlt | FedSim |
| Input Layer | **24.96**$_{0.01}$ | 24.44$_{0.01}$ | 24.81$_{0.01}$ | 51.97$_{0.02}$ | **53.94**$_{0.06}$ | 53.64$_{0.08}$ | 93.29$_{0.00}$ | **93.62**$_{0.03}$ | 93.55$_{0.05}$ |
| Output Layer | 24.93$_{0.01}$ | **25.05**$_{0.01}$ | 25.02$_{0.01}$ | 53.21$_{0.01}$ | **56.64**$_{0.05}$ | 56.24$_{0.04}$ | 93.37$_{0.01}$ | **93.57**$_{0.04}$ | **93.55**$_{0.05}$ |
| Adapter | 24.71$_{0.00}$ | **24.82**$_{0.01}$ | 24.74$_{0.01}$ | 63.86$_{0.06}$ | **66.41**$_{0.05}$ | 66.35$_{0.03}$ | 93.66$_{0.00}$ | **94.13**$_{0.03}$ | 94.07$_{0.03}$ |

StackOverflow/GLDv2 and 10 per round for EMNIST, for both the non-personalized as well as the personalized training.

**Local Updates and Minibatch Size.** Each selected device locally runs 1 epoch of mini-batch stochastic gradient descent locally for non-personalized as well as personalized federated training. The final finetuning at the end of personalized training is performed for 5 epochs. We use a minibatch size of 64 for StackOverflow/GLDv2 and 32 for EMNIST for all settings.

**Server and Client Optimizer Details.** We use FedAvg for EMNIST and FedAdam (Reddi et al., 2021) for StackOverflow and GLDv2. We also use a global scheduler, which applies a schedule on the client learning rates across rounds, while the client learning rate within each round is held constant. We use either a linear scheduler or an exponential scheduler (also called "stepLR" in PyTorch). A linear scheduler applies a linear warmup, if applicable, until the maximum learning rate followed by a linear decay to 0. An exponential scheduler halves the client learning rate once every fixed number of rounds. Both the client and server learning rates are tuned using the validation set.

**Regularization Coefficient for pFedMe and Ditto.** We tune the regularization coefficient $\lambda_k = \lambda$ for pFedMe and Ditto using the validation data from the set $\{10^{-4}, 10^{-3}, \cdots, 10^0\}$ of possible values. The tuned values are:

- StackOverflow: $10^{-3}$ for Ditto and $10^{-4}$ for pFedMe,
- GLDv2: $10^{-1}$ for both Ditto and pFedMe,
- EMNIST: $10^{-1}$ for both Ditto and pFedMe.

**Random Seed.** We report numbers averaged over 5 random seeds.

## C  EXPERIMENTS: ADDITIONAL RESULTS

We now present the detailed experimental results.

We start with Table 9, which shows the standard deviations of Table 4. We see that the numbers are fairly consistent across runs so the standard deviation is quite small (0.01 to 0.05 percentage points).

### C.1  ABLATION: FINAL FINETUNING FOR FEDALT AND FEDSIM

We now study the effect of the final finetuning (step (c) of the experimental pipeline; cf. §B.2) for FedAlt and FedSim.

**The final finetuning has a minimal impact on partial personalization.** We see from Table 10 that the effect of the final finetuning is much smaller than the improvements from personalization. For instance, the improvements from finetuning are close to 0 for FedAlt on the StackOverflow dataset. For GLDv2, the finetuning accounts for $< 0.5$pp of improvement, whereas personalization overall accounts for 5 to 15pp.

**The final finetuning is more important to FedSim than FedAlt.** Table 10 also shows that the final finetuning helps FedSim more than FedAlt. However, FedAlt still outperforms FedSim, as we saw in

Table 10: The change in accuracy (percentage points) from the final finetuning for FedAlt and FedSim with stateful devices. The subscript denotes the standard deviation over 5 random seeds.

| | StackOverflow | | GLDv2 | | EMNIST | |
|---|---|---|---|---|---|---|
| | FedAlt | FedSim | FedAlt | FedSim | FedAlt | FedSim |
| Input Layer | $-0.06_{0.01}$ | $0.04_{0.02}$ | $0.12_{0.02}$ | $0.17_{0.03}$ | $0.12_{0.01}$ | $0.12_{0.03}$ |
| Output Layer | $0.00_{0.01}$ | $0.25_{0.02}$ | $0.49_{0.02}$ | $0.57_{0.03}$ | $0.09_{0.01}$ | $0.09_{0.03}$ |
| Adapter | $0.01_{0.01}$ | $0.40_{0.08}$ | $0.14_{0.02}$ | $0.17_{0.01}$ | $0.27_{0.02}$ | $0.33_{0.03}$ |

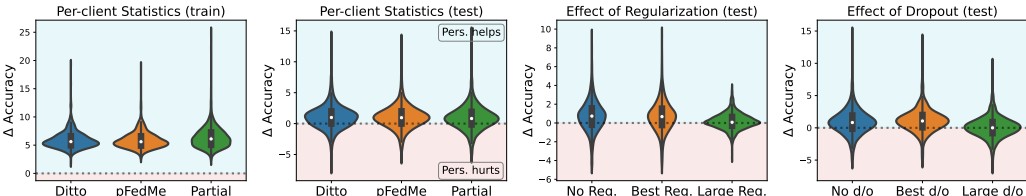

Figure 6: **Left two**: Distribution of change in the per-device train (left most) and test (center left) accuracy due to personalization on the StackOverflow dataset. **Right two**: Distribution of change in the per-device test accuracy of partial personalization under regularization on the StackOverflow dataset: (a) center right: adapter personalization under $\ell_2$ regularization, and, (b) rightmost: output layer personalization under dropout. Note that the "No Reg." and "No d/o" plots on the right two are different because they personalize different model parts. **Interpretation**: The white dot in inside the violin denotes the median, while the black box enclosing this white dot marks the interquartile range (i.e., $25^{\text{th}}$ and $75^{\text{th}}$ percentiles). The body of the violin is a kernel density estimate of the distribution of accuracies. The lines extend out to the minimum and maximum accuracy in each case.

Table 9. Overall, this shows that FedAlt is a better algorithm than FedSim. The final finetuning helps FedSim make up some percentage points in accuracy, but not enough to make up its gap with FedAlt.

### C.2 EFFECT OF PERSONALIZATION ON PER-DEVICE GENERALIZATION

**Summary of all scatter plots.** All the scatter plots shown in the main paper are summarized in the violin plot of Figure 6. We see from the leftmost figure that the training accuracies on all devices improve with personalization. From the second figure, we see that the test accuracy of some of the devices reduces with personalization; this is true for both partial and full personalization.

From the third plot of Figure 6, we see that regularization does not mitigate this overfitting. In fact, the regularization tuned for best average accuracy leads to a nearly identical distribution of test accuracies. A larger regularization reduces the spread of accuracies, but does so at the expense of a smaller median (white dot). The fourth plot of Figure 6 shows that the effect of dropout is similar. The best dropout improves the median accuracy, but it does not mitigate the issue of some devices being hurt by personalization.

**Train Accuracy plots for devices.** From Figure 7, we see that personalization leads to *a reduction in test accuracy* on some of the devices beyond the initial non-personalized model. The corresponding train accuracy plot is given in Figure 7. We observe that the personalization always leads to an improvement in the training accuracy but not in the test accuracy. The analogous plots for GLDv2 are in Figure 8, where the trends are similar.

**Whether personalization helps a device or not depends on the random seed.** We see in Figure 10 that the shaded region for some of the devices intersects the dotted line at $0$. In other words, personalization sometimes helps this device and sometimes hurts it, depending on the random seed. This indicates that the best fix in practice is to use A/B testing on the *deployed model* to choose whether to use the personalized model or the non-personalized one.

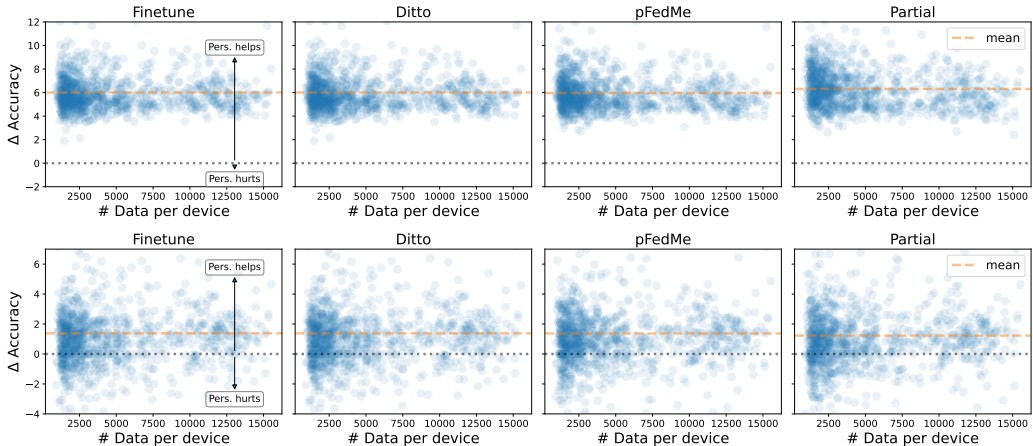

Figure 7: Scatter plot of change in accuracy (pp) per-device versus the number of training samples on the device for StackOverflow. **Top**: Training accuracy. **Bottom**: Test accuracy. This is the full version of Figure 4 from the main paper.

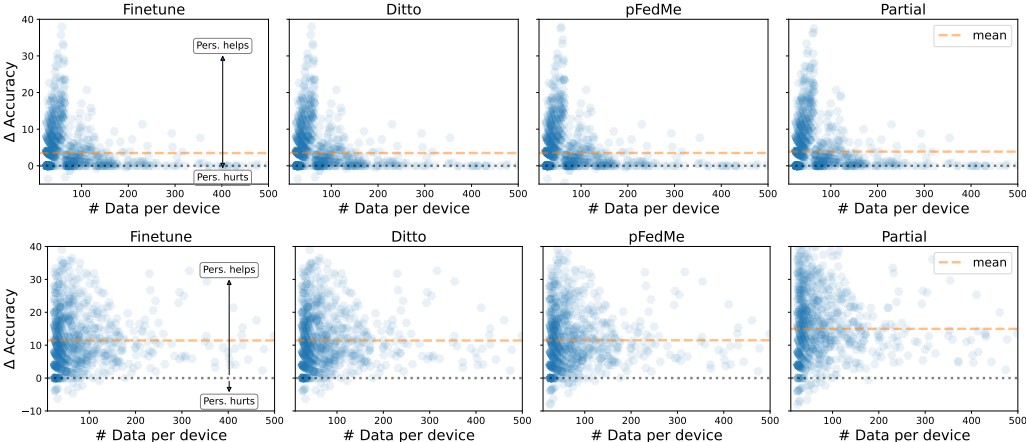

Figure 8: Scatter plot of change in accuracy (pp) per-device versus the number of training samples on the device for GLDv2. **Top**: Training accuracy. **Bottom**: Test accuracy.

**Regularization and dropout do not mitigate this issue.** From the first row of Figure 9, we see that the weight decay with best mean accuracy exactly matches the unreguarlized case in terms of per-device statistics. Increasing the regularization weight can reduce the spread of per-device accuracy. However, this only leads to a worse mean accuracy and does not mitigate the issue of personalization hurting individual devices.

From the second row of Figure 9, we see that the best dropout (0.3 in this case) leads to slight increase in average accuracy (0.18 pp). It also reduces the number of devices hurt by personalization from 256 out of 1000 to 193, but it does not fix this issue. Increasing dropout further only leads to a degradation of per-device statistics.

### C.3 PARTIAL PERSONALIZATION FOR STATELESS DEVICES

The algorithms we considered in this paper, namely FedAlt and FedSim, require the devices to maintain the personalized parameters $v_i$'s as state across rounds. In cross-device federated learning settings, it is also interesting to consider *stateless* devices, which are not allowed to maintain state between training rounds.

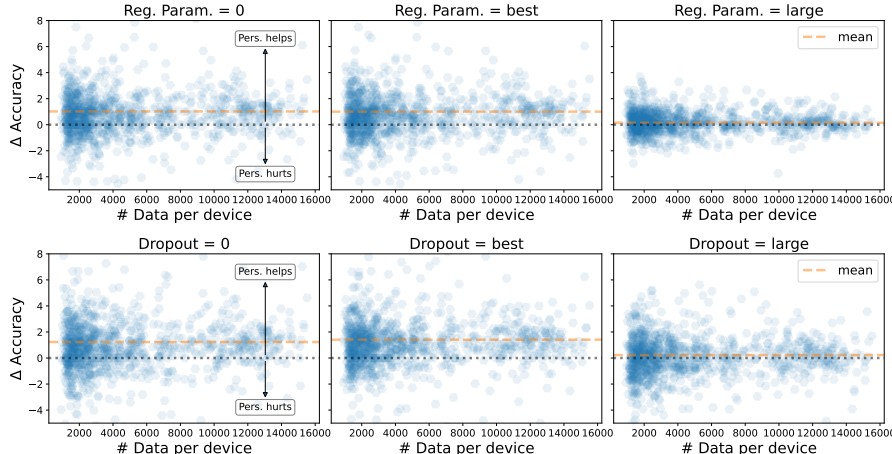

Figure 9: Scatter plot of change in accuracy (pp) per-device versus the number of training samples on the device with the effect of regularization. **Top**: $\ell_2$ regularization a.k.a. weight decay. **Bottom**: dropout. The "best" values of the $\ell_2$ regularization parameter and dropout are chosen to maximize the average test accuracy across all devices.

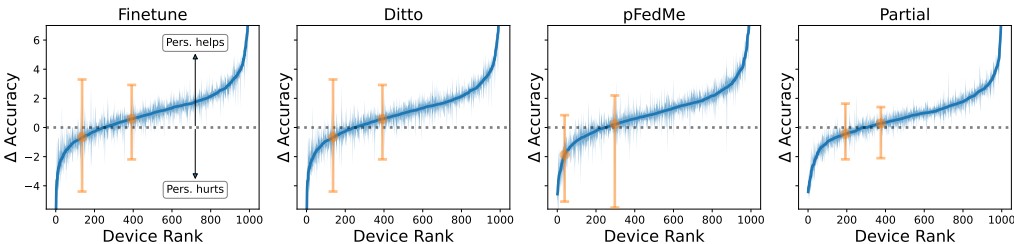

Figure 10: Change in per-device accuracy (pp) due to personalization. The solid line is the mean over 5 random runs and the shaded area denotes the max/min across these runs. The devices are sorted in ascending order of accuracy change. The points in orange depict two example devices who might either be helped or harmed by personalization depending on the random seed.

We give preliminary experiments in this setting. We modify the FedAlt and FedSim algorithms from the main paper so that the personalized parameters $v_i$ are reinitialized each time device $i$ is chosen for participation. We warm-start $v_i$ from the appropriate part of the non-personalized model trained in step (a) of the pipeline. For adapters, we fix a random initialization once, and reuse it.

**FedAlt is better than FedSim for stateless devices, although the improvement is smaller.** We see from Table 11 that all algorithms perform similarly for the stateless setting. Nevertheless, we see that FedAlt obtains mild improvements over both FedSim and finetuning for GLDv2, e.g., $0.24$pp with adapters.

**The final finetuning is crucial for stateless devices.** We see from Table 12 that the final finetuning accounts for most of improvements in the stateless case. For instance, for GLDv2, the final finetuning accounts for $11.68$ and $10.42$pp out of a total of $12.67$ and $11.76$pp for FedAlt and FedSim respectively. However, the personalized federated training (step (b) of the pipeline; cf. §B.2) still leads to an increase in accuracy of $1$ to $1.34$pp.

Table 11: This is the counterpart of Table 4 to stateless devices. We compare FedAlt and FedSim for partial model personalization with stateless devices. "FT (part.)" corresponds to finetuning the personal parameters $v_i$ locally while fixing the shared parameters $u$ from a non-personalized training. The numbers are averaged over 5 random seeds; the boldfaced numbers denote the highest accuracy in each row.

| | StackOverflow | | | GLDv2 | | | EMNIST | | |
|---|---|---|---|---|---|---|---|---|---|
| | FT (part.) | FedAlt | FedSim | FT (part.) | FedAlt | FedSim | FT (part.) | FedAlt | FedSim |
| Input Layer | $\mathbf{24.96}_{0.01}$ | $24.84_{0.01}$ | $24.89_{0.01}$ | $51.97_{0.02}$ | $\mathbf{52.76}_{0.06}$ | $52.74_{0.02}$ | $93.29_{0.00}$ | $\mathbf{93.51}_{0.03}$ | $93.48_{0.04}$ |
| Output Layer | $24.93_{0.01}$ | $\mathbf{24.94}_{0.01}$ | $24.94_{0.01}$ | $53.21_{0.01}$ | $\mathbf{53.30}_{0.06}$ | $53.30_{0.08}$ | $93.37_{0.01}$ | $\mathbf{93.53}_{0.03}$ | $93.51_{0.04}$ |
| Adapter | $\mathbf{24.71}_{0.00}$ | $24.69_{0.01}$ | $24.71_{0.01}$ | $63.86_{0.06}$ | $\mathbf{64.10}_{0.14}$ | $63.19_{0.04}$ | $93.66_{0.00}$ | $\mathbf{93.97}_{0.04}$ | $93.89_{0.02}$ |

Table 12: The change in accuracy (percentage points) from the final finetuning for FedAlt and FedSim with stateless devices. The subscript denotes the standard deviation over 5 random seeds.

| | StackOverflow | | GLDv2 | | EMNIST | |
|---|---|---|---|---|---|---|
| | FedAlt | FedSim | FedAlt | FedSim | FedAlt | FedSim |
| Input Layer | $0.86_{0.03}$ | $1.00_{0.02}$ | $0.44_{0.03}$ | $0.42_{0.03}$ | $0.11_{0.02}$ | $0.10_{0.04}$ |
| Output Layer | $1.08_{0.03}$ | $1.10_{0.02}$ | $1.47_{0.04}$ | $1.46_{0.05}$ | $0.15_{0.02}$ | $0.11_{0.02}$ |
| Adapter | $0.84_{0.04}$ | $0.88_{0.02}$ | $11.68_{0.20}$ | $10.42_{0.09}$ | $0.46_{0.02}$ | $0.42_{0.04}$ |

