# OpenReview forum: "Federated Learning with Partial Model Personalization"
_ICLR.cc/2022/Conference — ICLR 2022 Submitted_

### Official Review · Reviewer_8i9S · 2021-10-22

**Correctness:** 4
**Technical Novelty And Significance:** 2
**Empirical Novelty And Significance:** 2
**Recommendation:** 3
**Confidence:** 4

**Main Review:**

My main concern about the paper is the limited novelty. The idea of partial personalization has been proposed in previous literature. The objective function (3) has been detailly studied in [1]. Moreover, the convergence of FedSim for nonconvex case has also been studied in [1] under very similar assumptions (See Section 3.1 of [1], where the name of FedSim is LSGD-PFL). Thus, the only new result left is the analysis of FedAlt under nonconvex setting. In order to derive Theorem 2, the authors claim that they proposed a new technique called virtual full participation; however, this technique is actually not new and has been used widely in convergence analysis for FedAvg/Local-SGD. This way, the whole contribution of the paper relies on the analysis of FedAlt in nonconvex case based on well-known techniques, which seems quite weak contribution to me. I would recommend the authors to give more detailed explanation on what the key theoretical challenge in deriving Theorem 2 is.

The extensive experiments of using different model structures on different datastes is appreciated. And the observation personalization can hurt the test accuracy on some devices is interesting. But given the idea of partial personalization is proposed elsewhere, the gain from the experiments is limited.

The writing is clear. I can easily understand the paper. Thanks for that.

In a word, given the ideas used in the paper are now new, I think this paper is maybe better written in a pure theoretical one. The key would be to stress the theoretical challenge and novelty. Based on the current version, I believe the contribution is not novel enough or at least it is not stated clearly.

[1] Filip Hanzely, Boxin Zhao, and Mladen Kolar. Personalized Federated Learning: A Unified Framework and Universal Optimization Techniques. arXiv Preprint, 2021.

**Summary Of The Paper:**

The paper discusses using partial personalization objective function defined in equation (3) to achieve personalized models in federated learning. While the idea of partial personalization and the objective of (3) have been widely studied in previous literatures, the main contributions of the paper are the convergence analysis of two proposed algorithms FedSim and FedAlt in nonconvex case. The authors also did extensive experiments to compare partial personalized models with fully personalized models by using various model structures and on different datasets.

**Summary Of The Review:**

The contribution is unclear, which constitutes strong reason for not accepting the paper.

---

> ### Author Response · Authors · 2021-11-13
> **Reponse to reviewer 819S**
>
> We thank Reviewer 8i9S for their careful reading of the paper, and for appreciating the depth of our experimental evaluation. We address the concern over novelty below. We are happy to address any other questions you might have.
>
> **“The idea of partial personalization has been proposed in previous literature.”**
>
> Previous works have proposed personalization of a fixed layer e.g., input layer or output layer, while we give a general framework and advocate for the more general architecture where personalized versus shared partitions can be arbitrary and determined by domain expertise. Indeed, our experiments show that a fixed choice as in previous works can be suboptimal. For the GLDv2 dataset, we propose the use of adapters for personalization, and this outperforms previous approaches by 9.8pp (66.4% versus 56.6% top-1 accuracy).
>
> **“Objective function has been studied by Hanzely et al. (2021); they analyze FedSim too”**
>
> We cite Hanzely et al. for their unified treatment of various objectives and convergence analysis in the convex case. We had, however, missed their nonconvex analysis of FedSim -- thank you for bringing this to our attention. Upon closer inspection, Hanzely et al. analyze FedSim with full participation of clients in every round, while our analysis accounts for client sampling, hence addressing a more practical setting. We have updated the related work section on page 3 with a more precise comparison.
>
> **“virtual full participation; however, this technique is actually not new ...”**
>
> Our technique of virtual full participation is novel when compared to the standard technique of shadow iterates used in the Local SGD literature (e.g., Wang et al. 2021, Koloskova et al. 2020). We first explain our technique and then argue how it is different from that of previous literature.
>
> **Summary of virtual full participation**:
> 1. The updated personal parameters $v_i^{(t+1)}$ depend on the sampling of devices $S^{(t)}$, and the update of $u^{(t+1)}$ in turn depends on $v_i^{(t+1)}$. Therefore, when taking expectations w.r.t. $S^{(t)}$ in the $u$-step, we cannot get useful reductions due to the interdependency with $v_i^{(t+1)}$. Note that this problem only arises with client sampling (not with full participation) and only with FedAlt (not with FedSim).
> 2. We propose to analyze the $v$-step assuming *all* devices, including those not selected for updates, have performed $u$-steps on their personal variables. This removes the dependence on $S^{(t)}$. Then we show the progress made with partial participation is a fraction (proportional to number of participating devices) of the virtual case of full participation.
> 3. Then, through a careful error analysis, we bound the error caused by virtual full participation (i.e., these extra $u$-updates which actually did not happen): the difference is a lower order term $O(\gamma_u^2 \gamma_v^2)$; this can be controlled by making the learning rates small enough.
>
> Please see pages 23 and 24 for details on steps 1-2 and the last paragraph of page 25 for 3.
>
> **Novelty over standard techniques**:
> - As far as we know, shadow iterates (e.g., Wang et al. 2021, Koloskova et al. 2020) are only used with full client participation without personalization, and there they are used to quantify the effect of client drift due to local steps. Their construction is only well-defined in this case, while our technique is tailored to handling client sampling of the personal parameters. Notably, other work analyzing the case of client sampling without personalization (e.g., Karimireddi et al. 2020) does not use shadow iterates.
> - In our notation, the shadow sequence of Wang et al. is $\bar v_{k}^{(t)} = (1/n)\sum_i v_{i, k}^{(t)}$. This is different from our technique of virtual full participation because: (a) shadow iterates are well-defined only for full participation, and, (b) shadow iterates are defined for the shared parameters $v$, rather than the personal parameters $u_i$.
>
>
> **“... recommend … more detailed explanation on what the key theoretical challenge in deriving Theorem 2 is”**
>
> Thank you for your suggestion; we will outline the above argument more carefully in the next revision.

---

### Official Review · Reviewer_447D · 2021-10-31

**Correctness:** 4
**Technical Novelty And Significance:** 2
**Empirical Novelty And Significance:** 2
**Recommendation:** 5
**Confidence:** 4

**Main Review:**

Pros:
1. The authors proposed a unified framework for personalized federated learning. To my best knowledge, they give the first analysis of this framework and algorithm on smooth nonconvex function.
2. The proposed algorithm can outperform existing personalization algorithms on NLP or Vision tasks.

Cons:
1. The personalized layer idea is not new, as authors mentioned, it has been proposed in prior works [Arivazhagan et al. (2019) and Collins et al. (2021)].
2. I think the established theory does not perfectly fit with experiment model, since they assume smoothness in theory, but in practice the motivated example is usually non-smooth ReLU network. Hence I think it will be great if they can perform analysis on ReLU network, perhaps very simple one layer setting.



**Summary Of The Paper:**

This paper proposed a personalized FL framework with partial model personalization. It separates the model parameters into two parts, shared model and personalized model, and optimize them in an interleaving manner. The authors proposed two optimization algorithm, named as FedSim and FedAlt, with partial clients participation. They also analyzed the algorithms' convergence rate on general smooth nonconvex function. In experiments, they consider 3 different model split methods: 1. input layer as personalized model, the rest as shared model; 2. output layer as personalized model, the rest as shared model; 3. adding adapter as personalized model.  The experiments conduct on NLP or vision tasks also demonstrate that the proposed algorithm outperforms other personalization method.

**Summary Of The Review:**

Overall, this framework is interesting and theory is new. However, the novelty for me is somewhat limited. If they can come up with theory that can explain their empirical observations (ReLU network), I believe it will be a strong work.

---

> ### Author Response · Authors · 2021-11-13
> **Response to Reviewer 447D**
>
> Thank you for recognizing our theoretical and empirical contributions! We address the "cons" in detail below.
>
> **“The personalized layer idea is not new”**
>
> We propose a unified framework for federated learning with partial model personalization. It encompasses all the previous works on personalization layers. In addition, our framework allows for personalization with adapter modules, which empirically outperforms previously proposed personalization layers for image classification by 9.8pp on GLDv2 (66.4% versus 56.6% top-1 accuracy).
>
> **“established theory does not perfectly fit with experiment model”**
>
> Our theory only assumes that the expectation $F_i(w) = \mathbb{E}_{z \sim D}[f(w; z)]$ is smooth; see Assumption 2’ in Appendix A. This is possible even if the loss $f(w; z)$ is computed on a 2-layer ReLU network acting on data $z \sim D$ as long as the distribution $D$ is continuous without any atoms, such as a Gaussian distribution. See, for instance, Lemma 2 of [Nesterov (2011)](https://mipt.ru/dcam/upload/e04/RandomGradFree-arph2hev1iv.PDF) or Lemma 1.5 of [Abernethy et al. (2016)](https://dept.stat.lsa.umich.edu/~tewaria/research/abernethy16perturbation.pdf ).
>
> In addition, we have done extensive comparisons of ReLU and smoothed ReLU activations (e.g., softplus, ELU) in our previous works, and they do not show any significant difference.

---

### Official Review · Reviewer_Nawz · 2021-11-03

**Correctness:** 4
**Technical Novelty And Significance:** 2
**Empirical Novelty And Significance:** 2
**Recommendation:** 3
**Confidence:** 4

**Main Review:**

This is a well written paper and easy to follow. However, I have the following concerns.

An important related work is missing, and I strongly suggest the authors discuss
- [Singhal et al. 2021 Federated Reconstruction: Partially Local Federated Learning https://arxiv.org/abs/2102.03448]. If I understand correctly, the proposed FedAlt algorithm is very similar to the paper, except that FedAlt requires a local state v_i.

Clarification on contributions
If I understand correctly, FedSim has been used in [Liang et al. 2019, Arivazhagan et al. 2019, Collins et al. 2021, Li et al. 2021 ], FedAlt is relatively new but closely related to [Singhal et al. 2021]. This paper makes the following contributions
- Convergence guarantees of FedSim and FedAlt: standard convergence rate under reasonable assumptions.  Li et al. 2021 presented a convergence rate of FedSim, could the authors compare?  Regarding the novelty of FedAlt proof, I believe virtual proxy variable is a common technique in federated optimization, (e.g., eq(13) in [Wang et al. 2021 A Field Guide to Federated Optimization https://arxiv.org/pdf/2107.06917.pdf]), and I would appreciate it if the authors elaborate more on the novelty.
- FedAlt algorithm, which is a client state variant of Singhal et al. 2021, and achieves marginal improvement over FedSim [Liang et al. 2019, Arivazhagan et al. 2019, Collins et al. 2021, Li et al. 2021] (always < 0.5%.)
- Theoretical insights on when FedAlt is better than FedSim. I found this to be interesting, and I would encourage the authors to provide more discussion and connections to empirical results.


Savings of memory footprint
- It would be good to show either analytically or empirically how much memory the partial model training can reduce compared to full model training as this is the main motivation.

Clarification on some details.
-  Authors mentioned “All methods, including FedSim, FedAlt and the baselines are initialized with a global model trained with FedAvg. ”: how is the global model trained? And is it trained on the same set of clients as personalization?
- For table 3, does the full model in baselines include adapter parameters?
- How do the authors split the dataset for training, personalization and testing?


**Summary Of The Paper:**

This paper studies personalization in federated setting, i.e., instead of collaboratively training a global model, personalizing a model for each client.  This paper proposed to personalize only part of the model parameters instead of the full model, and studied two algorithms FedSim and FedAlt. In local client updates, FedSim will simultaneously train the shared and personalized parameters, while FedAlt will train the personalized model first, then train the shared parameter.

**Summary Of The Review:**

I strongly encourage the authors clarify contributions explicitly acknowledging previous works [Liang et al. 2019, Arivazhagan et al. 2019, Collins et al. 2021, Li et al. 2021 ], as well as  [Singhal et al. 2021 Federated Reconstruction: Partially Local Federated Learning https://arxiv.org/abs/2102.03448].

---

> ### Author Response · Authors · 2021-11-13
> **Response to Reviewer Nawz (1/2)**
>
> Thank you for the review and constructive comments! We address your comments below.
>
> **“An important related work is missing … Singhal et al. 2021”**
>
> Thank you for pointing out this key missing reference! We have updated the paper with this reference. Indeed, FedAlt is a variant of the algorithm of Singhal et al. with $v_i$’s maintained as state. Compared to Singhal et al., we provide a theoretical analysis of FedAlt, overcoming major technical challenges posed by client sampling (more on this below).
>
> **“... clarify contributions explicitly acknowledging previous works ...”**
>
> We do not claim novelty of proposing FedAlt or FedSim. To be precise, in the related work section on page 3, we had originally attributed FedSim to Liang et al. and Arivazhagan et al., and FedAlt to Collins et al. In the revision, we also attribute FedAlt to Singhal et al.
>
> Our contribution is the analysis of these algorithms in the general nonconvex case with client sampling. Please note that none of these previous works contain a theoretical analysis in this case, with the exception of Collins et al. for 2-layer linear networks in FedAlt and Hanzely et al. for FedSim with *full* participation. The key technical challenge we address is the analysis of FedAlt with client sampling (more on this below).
>
> On the modeling side, our framework allows personalizing arbitrary parts of the model, which are usually determined by domain expertise. Indeed, our experiments show that a fixed choice as in previous works can be suboptimal. For the GLDv2 dataset, we propose the use of adapters for personalization, and this outperforms previous approaches by 9.8pp (66.4% versus 56.6% top-1 accuracy).
>
> **“Regarding the novelty of FedAlt proof, I believe virtual proxy variable is a common technique”**
>
> Our technique of virtual full participation is novel when compared to the standard technique of shadow iterates used in the Local SGD literature (e.g., Wang et al. 2021). We first explain the technique and then argue how it is different from that of Wang et al.
>
> **Summary of virtual full participation**:
> 1. The updated personal parameters $v_i^{(t+1)}$ depend on the sampling of devices $S^{(t)}$, and the update of $u^{(t+1)}$ in turn depends on $v_i^{(t+1)}$. Therefore, when taking expectations w.r.t. $S^{(t)}$ in the $u$-step, we cannot get useful reductions due to the interdependency with $v_i^{(t+1)}$. Note that this problem only arises with client sampling (not with full participation) and only with FedAlt (not with FedSim).
> 2. We propose to analyze the $v$-step assuming *all* devices, including those not selected for updates, have performed $u$-steps on their personal variables. This removes the dependence on $S^{(t)}$. Then we show the progress made with partial participation is a fraction (proportional to number of participating devices) of the virtual case of full participation.
> 3. Then, through a careful error analysis, we bound the error caused by virtual full participation (i.e., these extra $u$-updates which actually did not happen): the difference is a lower order term $O(\gamma_u^2 \gamma_v^2)$; this can be controlled by making the learning rates small enough.
>
> Please see pages 23 and 24 for details on steps 1-2 and the last paragraph of page 25 for 3.
>
> **Novelty over standard techniques**:
> - As far as we know, shadow iterates (e.g., Wang et al. 2021, Koloskova et al. 2020) are only used with full client participation without personalization, and there they are used to quantify the effect of client drift due to local steps. Their construction is only well-defined in this case, while our technique is tailored to handling client sampling of the personal parameters. Notably, other work analyzing the case of client sampling without personalization (e.g., Karimireddi et al. 2020) does not use shadow iterates.
> - In our notation, the shadow sequence of Wang et al. is $\bar v_{k}^{(t)} = (1/n)\sum_i v_{i, k}^{(t)}$. This is different from our technique of virtual full participation because: (a) shadow iterates are well-defined only for full participation, and, (b) shadow iterates are defined for the shared parameters $v$, rather than the personal parameters $u_i$.
>
>
> **“Li et al. 2021 presented a convergence rate of FedSim”**
>
> Li et al. only prove convergence in the strongly convex case with *full* model personalization. We prove a rate in the general non-convex case with *partial* model personalization, applicable to deep learning. Further, it is not possible to directly apply or extend the analysis of Li et al. to the setting we consider because the shared parameter $v$ does not depend on the personal parameter $u_i$ in the work of Li et al. This dependence necessitates the use of virtual full participation in our case.

---

> > ### Author Response · Authors · 2021-11-13
> > **Response to Reviewer Nawz (2/2)**
> >
> > **“Difference between FedSim and FedAlt is marginal”**
> >
> > Our purpose is to provide convergence analysis and extensive empirical study of both algorithms, not to particularly emphasize which one is better. Our experiments show a small but consistent difference between FedSim and FedAlt, and this difference is much larger than the standard deviation across runs (see Table 9 for the standard deviation).
> >
> > **“insights on when FedAlt is better than FedSim”**
> >
> > We argue at the end of page 6 that FedAlt is less affected by the coupling $\chi$ between the personal and shared parameters. Here is some intuition on why this is so.
> >
> > Because we allow for client sampling (partial participation), when a client gets selected to perform an update, their personal parameters may have become stale. That is, the shared parameters would have been updated many times at the server since the personal parameters at that client were updated. By first updating the personal parameters, FedAlt ensures that the subsequent shared parameter updates returned to the server are more relevant.
> >
> > This insight is in line with FedSim and FedAlt being analogous to the classical Jacobi and Gauss-Seidel methods in linear algebra (see end of page 4).
> >
> > **“Savings of memory footprint”**
> >
> > Please see the table below for estimated peak memory usage in MB while training the models considered in the experiments.
> >
> > |   Mode                 | StackOverflow | GLDv2 | EMNIST |
> > |------------------------|---------------|-------|--------|
> > | No personalization     | 71            | 186   | 142    |
> > | Partial / Input layer  | 67            | 186   | 142    |
> > | Partial / Output layer | 67            | 174   | 142    |
> > | Partial / Adapter      | 72            | 222   | 159    |
> > | Full personalization   | 116           | 263   | 232    |
> > | Memory savings with partial personalization| **42%** | **34%** | **39%** |
> >
> > We estimate the memory footprint as follows:
> > - We assume that the following are needed to be stored on client during training: $v^{(t)}$ the previous broadcast global model, which is needed to calculate the model delta to be sent back to the server, current iterate of the shared parameter $v_{i,k}^{(t)}$, current iterate of the personal parameter $u_{i,k}^{(t)}$, the respective gradients $\nabla_v$ and $\nabla_u$ and the internal buffers required for backpropagation.
> > - The total memory consumption is therefore, $3 \times \text{size}(v) + 2 \times \text{size}(u) + \text{size(backprop)}$.
> >
> > **"Clarification on some details"**
> >
> > - The global model and personalized model are trained on the same set of clients and on the same training set. We use either FedAdam (StackOverflow and GLDv2) or FedAvg (EMNIST). Please see Table 8 and Section B.2 in the appendix for more details (number of pre-training rounds, learning rates and other hyperparameters).
> > - Each client has a fixed train-test split. The same training set is used for both non-personalized as well as personalized training. For StackOverflow and EMNIST, the train-test split is the canonical one given by TFF. For GLDv2, we take 50% of the data on each client as the test set.
> > - For table 3, the full model in the baseline does not include adapters.

---

> > > ### Comment · Reviewer_Nawz · 2021-11-18
> > > **More clarification**
> > >
> > > I have to check the theory again as the authors clarified the main contribution is in theory. I want to start a discussion now so that the authors still have a chance to update the draft.
> > >
> > > (1) Shadow iterates is commonly used, and I think the previous mentioned proof for FedAvg for full client participation can be extended to client sampling. A concrete example is Li et al. On the Convergence of FedAvg on Non-IID Data https://arxiv.org/abs/1907.02189.
> > >
> > > (2) The experimental results are less reliable as adapters are used for some models, but not the others. Models with adapters have more parameters.

---

### Official Review · Reviewer_mEwK · 2021-11-07

**Correctness:** 3
**Technical Novelty And Significance:** 2
**Empirical Novelty And Significance:** 2
**Recommendation:** 5
**Confidence:** 4

**Main Review:**

This paper studies an interesting question and proposed idea looks interesting with corroborating empirical results, but there are few issues that prevents me from giving it a high score:

- The key motivation for the paper is to utilize a small footprint in clients but the proposal is somehow mis-leading. I understand the learned personalized model is significantly smaller than the shared model (e.g. 1-2% as observed in experiments for some applications) and requires a small footprint after deployment (inference stage), but during the training the clients need enough memory for both models to participate in collaborative training. Also, I was left wondering how large the coupling between personalized and shared parameters captured by $\chi$ would be in presence of heterogeneity and significant difference in model sizes.

- While the proof of theoretical results looks sound as far as I checked, the obtained rates are hard to interpret and poorly elaborated. For example, the gradient diversity assumption (Assumption 3) is only defined over the shared model. Also, the role of gradient diversity term, \rho, which appears in tuning the learning rate $\eta$ is ignored in discussions (e.g. Table 1) which might dominate other terms if incorporated properly. So, it seems hard to put the obtained rates in the context of known results.


**Summary Of The Paper:**

To overcome statistical heterogeneity among data shards in collaborative federated learning, this paper proposes a novel personalization schema which only requires smaller memory footprint in clients and possibly is less susceptible to catastrophic forgetting. The main idea is to split the trainable parameters into  shared and personalized parameters where, unlike existing personalization schema, only the shared model is exchanged with the server in communication rounds to be aggregated- thus partial personalization. In regression, this corresponds to learning the residual error of shared model via personalized model and in the classification setting corresponds to output averaging (unlike interpolation based personalization methods that do parameter mixing). The authors propose two algorithms FedSim (simultaneous updating of shared and personalized models locally) and FedAlt (alternative updating of shared and personalized models locally), to learn in this setting, theoretically analyze the convergence rates in non-convex settings and conduct empirical studies on image classification and next-word prediction to evaluate the proposed methods.


**Summary Of The Review:**

This paper proposes a solution for personalization in federated learning where clients only exchange a small portion of parameters with a server for collaborative learning. The problem and proposal looks interesting, but it would be great to hear from authors on some issues raised above before final evaluation.

---

> ### Author Response · Authors · 2021-11-13
> **Response to Reviewer mEwK**
>
> Thank you for appreciating our theoretical and empirical contributions! We address in detail the issues you raised. We are happy to answer if you have any further questions.
>
> **“during the training the clients need enough memory for both models”**
>
> We claim that the memory footprint of partial personalization is much smaller than that of *full personalization* during training, and it is roughly the same as that of *no personalization*. During deployment, each method has a single full model on-device, so the memory consumption is the same. Please see the table below for estimated peak memory usage in MB while training the models considered in the experiments.
>
> |   Mode                 | StackOverflow | GLDv2 | EMNIST |
> |------------------------|---------------|-------|--------|
> | No personalization     | 71            | 186   | 142    |
> | Partial / Input layer  | 67            | 186   | 142    |
> | Partial / Output layer | 67            | 174   | 142    |
> | Partial / Adapter      | 72            | 222   | 159    |
> | Full personalization   | 116           | 263   | 232    |
> | Memory savings with partial personalization| 42% | 34% | 39% |
>
>
> **“how large (is) the coupling between personalized and shared parameters captured by $\chi$”**
>
> We work out $\chi$ in two simple cases. The first one is the generalized additive linear model of Fig. 2b, where sensitive features $x_2$ are handled locally, while less-sensitive features $x_1$ are handled by a shared model. The second one is the full personalization case of Eq. 2, and $f$ is assumed to be $L$-smooth. In both these cases, we have $\chi = O(1)$.
>
> | Example                          | $F(u, v)$                             | $\chi^2$                                 | Bound on $\chi^2$ |
> |----------------------------------|---------------------------------------|------------------------------------------|-------------------|
> | Additive model  (Fig. 2b)                       | $(u^\top x_1 + v^\top x_2 - y)^2$     | $(x_1^\top x_2)^2 / \Vert x_1\Vert^2 \Vert x_2\Vert^2$ | 1                 |
> | Full personalization (Eq. 2) | $f(v) + \frac{\lambda}{2}\Vert u - v\Vert^2$ | $\lambda^2 / \lambda(\lambda + L)$       | 1                 |
>
>
> **“the gradient diversity assumption (Assumption 3) is only defined over the shared model”**
>
> Our gradient diversity assumption is a natural analogue of that from federated learning without personalization e.g., Karimireddi et al. 2021. We only need the diversity assumption over the shared model, while the personalized parts can be arbitrarily heterogeneous.
>
> **“the role of gradient diversity term $\rho$ (...) is ignored in discussions”**
> The full rate of convergence is given in Corollary 3 (top of page 16) for FedSim and Corollary 10 (top of page 23) for FedAlt. The terms containing $\rho$ are lower order, so we omit them in the main paper for simplicity.

---

### Author Response · Authors · 2021-11-15
**Thank you for the reviews | Reiterating our contributions**

We thank the reviewers for their constructive comments and pointers to important references and missing citations. After addressing the comments of each reviewer, we would like to re-emphasize our contributions:

* **Convergence Analysis**: The convergence analysis of FedSim and FedAlt in the nonconvex setting with partial participation are new and novel. FedSim in the nonconvex setting with full participation has been analyzed by Hanzely et al. (2021), but the partial participation case is important for practical scenarios. The analysis of FedAlt with partial participation is especially challenging and requires novel techniques that we propose. They provide a general set of theoretical results to support personalized FL with nonconvex deep learning models.

* **Flexible Modeling**: As we acknowledged in the paper, personalizing different parts of the models have been proposed for different reasons and to suit different domain applications. But it is still a very meaningful contribution by providing a general framework that allows arbitrary partial model personalization, together with rigorous convergence guarantee in the general nonconvex setting. Our empirical evidence clearly shows the importance of flexibility of choosing the right part of the model for personalization.

* **Extensive Experiments and New Phenomena**: We conducted extensive empirical study on realistic real-world datasets that cover tasks in computer vision and natural language processing. These experiments give convincing evidence of the advantages of partial model personalization in terms of memory saving, overall performance improvement, as well as evidence of being less susceptible to catastrophic forgetting. Moreover, our results reveal some important phenomena that have been overlooked in previous work; specifically, personalization causes some clients to have worse performance despite the overall improvement, which calls for future research on robustness and fairness.

Overall, we believe these contributions are solid and comprehensive, and we hope the reviewers can consider these points in finalizing their evaluations.

---

### Decision · Program_Chairs · 2022-01-20

**Decision:**

Reject

**Comment:**

Dear authors,

I have read the reviewers and your careful rebuttals. I would have liked to see much more engagement from the reviewers. However, even after your rebuttal, no reviewer suggested acceptance, with two reviewers proposing reject (3) and two proposing weak reject (5).

The reviewers found the paper well written. I concur. The reviewers also notice that the contributions are very marginal compared to prior literature. Personalized FL formulation studied here was in a simpler form first proposed by Hanzely and Richtarik (Federated learning of a mixture of global and local models, 2020) and later generalized by Hanzely et al - a paper the authors cite. That work performed an in-depth analysis, also including the nonconvex case, which the authors (claim) did not notice. Compared to that work, the authors perform an analysis in the partial participation regime. However, partial participation is by now a standard technique which can usually be combined with other techniques without much difficulty. The authors tried to argue that their analysis approach is unique, but the reviewers remained unconvinced.

In summary, I think this is a solid piece of work which is perhaps judged, looking at the raw scores, a bit too harshly. However, most verbal comments are indeed fair. I am also of the opinion that the paper in its current form does not reach the necessary bar for acceptance. I would encourage the authors to carefully revise the manuscript, taking into account all feedback that they find useful. I think the paper can be improved, with not too much effort perhaps, to a state in which the bar could be reached.

Kind regards,

Area chair